# Policy Regret Minimization in Partially Observable Markov Games

## Abstract

We study policy regret minimization in partially observable Markov games (POMGs) between a learner and a strategic adaptive opponent who adapts to the learner's past strategies. We develop a model-based optimistic framework that operates on the learner-observable process using *joint* MLE confidence set and introduce an Observable Operator Model-based causal decomposition that disentangles the coupling between the world and the adversary model. Under multi-step weakly revealing observations and a bounded-memory, stationary and posterior-Lipschitz opponent, we prove an $\mathcal{O}(\sqrt{T})$ policy regret bound. This work advances regret analysis from Markov games to POMGs and provides the first policy regret guarantee under imperfect information against an adaptive opponent.

## 1 Introduction

Reinforcement learning (RL) has achieved remarkable empirical success across a wide range of challenging AI applications in recent decades (Mnih et al., 2015; Silver et al., 2016; 2017; Akkaya et al., 2019; Deepmind, 2024; Guo et al., 2025). Many of these problems can be naturally formulated as multi-agent reinforcement learning (MARL), where multiple learners interact in a dynamically evolving environment jointly influenced by other learning agents (Zhang et al., 2021).

In many applications of MARL, the learner interacts with *adaptive* players in an *asymmetric* setting, where the learner commits to a strategy at the beginning of each episode while the other agents subsequently adjust their strategies in response to pursue their own objectives. In addition, the learner is often required to make decisions despite lacking of complete information about the underlying states. For example, consider a simplified economic game between a government (the learner) and a population of companies (adaptive agents). The government announces tax policies, which are publicly observable, and subsequently collects tax revenues based on the companies' reported outcomes, while the companies adapt their strategies to maximize profit conditional on the announced policies (Zheng et al., 2020). Importantly, the government's information is inherently limited: firms' production costs, demand conditions, investment and R&D plans, as well as potential collusive behavior remain private and unobserved. Consequently, while the companies *adapt* their strategies based on the observed sequence of tax policies, the government must optimize under *partial observability* of the economic environment to achieve objectives such as maximizing social welfare.

Despite its prevalence, it remains largely unclear how to learn an optimal decision-making policy under partial observability when facing adaptive adversaries. Existing literature typically addresses adaptive adversaries and partial observability in isolation. For partial observability in multi-agent settings, Liu et al. (2022b) study the problem of learning toward various equilibria—such as Nash, Correlated Equilibrium, and Coarse Correlated Equilibrium—in Partially Observable Markov Games (POMGs), a natural generalization of Markov games to partially observable settings. However, their framework evaluates learning success only through external regret, which compares the learner's strategy sequence against the adversary's best response conditioned on that sequence. External regret, however, fails to capture the counterfactual nature of adaptive agents: it ignores how opponents might have responded differently had the learner followed an alternative strategy. To address this limitation, Nguyen-Tang & Arora (2025; 2024) initiated the study of learning against adaptive adversaries in Markov games under the notion of **policy regret** (Arora et al., 2012), which evaluates the learner's performance against the return they would have obtained by following an alternative policy, given the adaptivity of the opponent to the alternative policy. Nevertheless, these results do not extend to partial observability, a setting that is ubiquitous in MARL domains. It thus

remains an open question how a learner can make decisions against adaptive adversaries without full access to the underlying states.

In this paper, we develop the first unified theoretical and algorithmic framework for policy regret minimization in partially observable Markov games (POMGs). Since learning in POMGs is notoriously challenging—even in terms of external regret (Papadimitriou & Tsitsiklis, 1987)—we focus on a broad subclass of POMGs, namely weakly revealing POMGs, which are known to be tractable for external regret minimization (Liu et al., 2022c). The weakly revealing condition requires only that the joint observations of all agents disclose a nontrivial amount of information about the latent states, a property that is satisfied in many real-world applications. We show that, for such rich class of POMGs, under natural structural assumptions on the behavior of the adaptive adversaries, policy regret minimization is sample-efficient. In particular, our key technical contributions are as follows:

1. We identify a rich class of adaptive-adversary behaviors that allow sample-efficient policy regret minimization in multi-step weakly revealing POMGs. Our problem class is defined by the *novel posterior-Lipschitzness* condition (see Assumption 1.3), which constrains the adversary's posterior response, together with the Eluder condition on the world and adversary channel operators, arising from our *novel causal decomposition* of the Observable Operator Model (see Lemma 1.2).

2. We develop a unified algorithmic framework for policy regret minimization in weakly revealing POMGs (see Algorithm 1). Our framework combines the optimistic MLE approach of (Liu et al., 2023) with the mini-batch techniques of (Arora et al., 2012; Nguyen-Tang & Arora, 2024; 2025) in a novel way, enabling simultaneous learning of both the world model and the adversary model in multi-step weakly revealing POMGs.

3. For the proposed rich problem classes, we show that our unified algorithmic framework achieves a policy regret bound in the order of $\widetilde{\mathcal{O}}\left( H \left( m + \sqrt{d_C} \right) \sqrt{d_E T} \right)$, where $T$ is the number of episodes, $m$ is the adversary's memory, and $H$ is the horizon of the POMG. Here, $d_E$ denotes the joint Eluder dimension of the world and adversary model operators, capturing the intrinsic complexity of exploration, while $d_C$ is the log-covering number of the joint world–adversary model, measuring the richness of the overall model class. To the best of our knowledge, this is the first result establishing a sublinear policy regret bound for POMGs.

## 1.1 Overview of Techniques

Despite the modularity and simplicity of our algorithmic framework (and its apparent hindsight clarity), establishing our theoretical guarantees requires overcoming major technical challenges arising from the *coupled dynamics* of the world and adversary models. We address these challenges through the following key novel ideas.

- **Joint modeling via a *single* confidence set.** In POMGs, the learner observes only its own trajectory $\tau_A$, where the effects of the environment dynamics ($\theta$) and the opponent's strategy ($\Phi$) are entangled. This creates an identifiability problem: outcomes cannot be uniquely attributed to either stochasticity in the environment or the opponent's choices. Thus, maintaining separate confidence sets for $\theta$ and $\Phi$ is fundamentally unsound. To address this, our proposed algorithm (see Algorithm 1) maintains a single joint confidence set $\mathcal{C} \subseteq \Xi$ over the full system parameter $\xi = (\theta, \Phi)$.

- **Causal separation of world and wdversary models in the Observable Operator Model framework.** Building on the Observable Operator Model (OOM) results of (Liu et al., 2022a), our Stackelberg setting introduces a challenge that their techniques cannot address. Specifically, each per-step operator $J_h^{\xi,\pi}$ is a *coupled* black box, jointly dependent on $\xi = (\theta, \Phi)$. Our main technical contribution is to prove that each operator $J_h^{\xi,\pi}$ admits a factorization $J_h^{\xi,\pi} = G_h^{\Phi,\pi} \circ W_h^{\theta}$, where $W_h^{\theta}$ depends only on the environment and $G_h^{\Phi,\pi}$ only on the adversary and the learner's policy, thereby disentangling their effects in the OOM analysis (Lemma 1).

- **Reduction from Stackelberg POMGs to an augmented POMDP.** A key step in our analysis is reducing adaptive-adversary Stackelberg POMGs to an augmented POMDP with state $s_h' = (s_h, \zeta, \tau_{A,h-1})$. Together with the causal decomposition introduced above and the mini-batched design of (Nguyen-Tang & Arora, 2025), this reduction enables the application of Observable Operator Model (OOM) tools under $\alpha$-weakly revealing observations. In turn, this yields a solution to policy regret minimization in weakly revealing POMGs (Theorem 1).

## 1.2 RELATED WORK

**Policy regret minimization in MARL.** Policy regret has been widely used to analyze learning against adaptive adversaries in online learning (Arora et al., 2012) and repeated games (Arora et al., 2018), and has only recently been extended to multi-agent RL. Existing results, however, are limited to fully observable Markov games. In particular, Nguyen-Tang & Arora (2024) initiated the study of policy regret in Markov games, establishing fundamental barriers and providing sufficient conditions for achieving sublinear policy regret in tabular settings. Subsequently, (Nguyen-Tang & Arora, 2025) extended these results to Markov games with function approximation.

**Partially observable Markov games (POMG).** POMGs provide a general framework for modeling multi-agent sequential decision-making under uncertainty, extending single-agent POMDPs to settings with multiple agents, each with their own partial perspective and objectives. Early work by (Hansen et al., 2004) laid the foundational formalism for POMGs and explored dynamic programming solutions, though scalability and sample efficiency remain significant challenges. Recent research has sought to address these limitations directly; for instance, Liu et al. (2022b) investigate sample-efficient reinforcement learning for weakly revealing POMGs, providing theoretical guarantees for learning to minimize the external regret in this setting. Alongside these general advances, a substantial thread of literature has focused on finding equilibrium solutions, often under simplifying assumptions such as myopic follower behavior (Zhong et al., 2021) or complete information settings (Gerstgrasser & Parkes, 2023). The field has also seen a growing integration with deep reinforcement learning, with algorithms like Multi-Agent PPO (MAPPO) (Lowe et al., 2017) enabling empirical progress in complex environments. Brero et al. (2022) introduces the Stackelberg POMDP, a reinforcement learning framework for economic design that models the interaction between a mechanism designer (leader) and strategic participants (followers) as a Stackelberg game.

## 2 PROBLEM SETUP AND PRELIMINARIES

We study two-player general-sum partially observable Markov games (POMGs) (Hansen et al., 2004) in a tabular, episodic setting, which is fully specified by the tuple: $\mathcal{M} = (H, \mathcal{S}, \mathcal{A}, \mathcal{B}, \mathcal{O}_A, \mathcal{O}_B, \mathbf{T}, \mathbf{E}, \rho_0, r_A, r_B)$, where $H \in \mathbb{N}$ is the horizon; the latent state space is $\mathcal{S}$ with $|\mathcal{S}| = S$; the learner (player $A$) and the opponent (player $B$) act in $\mathcal{A}$ and $\mathcal{B}$ with $|\mathcal{A}| = A$, $|\mathcal{B}| = B$; the individual observation spaces are $\mathcal{O}_A$ and $\mathcal{O}_B$ with $|\mathcal{O}_A| = O_A$, $|\mathcal{O}_B| = O_B$. Let $\mathcal{O} := \mathcal{O}_A \times \mathcal{O}_B$ denote the joint observation at step $h$ by $o_h = (o_{A,h}, o_{B,h}) \in \mathcal{O}$. The controlled dynamics are given by the transition kernels $\mathbf{T}_h(\cdot \mid s, a, b) \in \Delta_{\mathcal{S}}, \forall h \in [H]$, and the emission kernels $\mathbf{E}_h(\cdot \mid s) \in \Delta_{\mathcal{O}}, \forall h \in [H]$. The initial state is sampled from $\rho_0 \in \Delta_{\mathcal{S}}$. Rewards are bounded and, for notational simplicity, depend only on local observations: for $i \in \{A, B\}$ and $h \in [H]$, $r_{i,h} : \mathcal{O}_i \to [0, 1]$. This specification covers cooperative, competitive (including zero-sum), and mixed-motive interactions through the independent reward functions $(r_A, r_B)$.

**Interaction protocol.** An episode starts with a random initial state $s_1 \sim \rho_0$. At every step $h$ within the episode, a joint private observation $o_h = (o_{A,h}, o_{B,h}) \sim \mathbf{E}_h(\cdot \mid s_h)$ is drawn from the emission kernel $\mathbf{E}_h$ conditioned on the current latent state $s_h$. The learner (respecitvely, the opponent) selects an action $a_h$ (respectively, $b_h$) based on her respective private per-episode history $\tau_{A,h} = (o_{A,1}, a_{A,1}, \ldots, o_{A,h})$ (respectively, $\tau_{B,h} = (o_{B,1}, a_{B,1}, \ldots, o_{B,h})$). Note that in a POMG, states are hidden from all the players and each player $i \in \{A, B\}$ observes only her own historay $\tau_{i,h}$. The episode termnates after $H$ steps.

**Policies and value functions.** A policy $\pi = (\pi_1, \ldots, \pi_h)$ for the learner is defined as a map: $\pi_h : (\mathcal{O}_A \times \mathcal{A})^{h-1} \times \mathcal{O}_A \to \Delta(\mathcal{A})$, for all $h \in [H]$, where $\Delta(\mathcal{A})$ is the set of all distributions over $\mathcal{A}$. A policy $\mu = (\mu_1, \ldots, \mu_H)$ for the adversary is defined similarly: $\mu_h : (\mathcal{O}_B \times \mathcal{B})^{h-1} \times \mathcal{O}_B \to \Delta(\mathcal{B}), \forall h \in [H]$. We assume that the learner and the adversary select their policies from a *restricted* class of policies, $\Pi$ and $\Psi$, respectively.

The world model $\theta = (\mathbf{T}, \mathbf{E}) \in \Theta$ characterizes the POMG with a transition kernel $\mathbf{T}$ and an emission $\mathbf{E}$. Let $\tau = \{(\tau_{A,h}, \tau_{B,h})\}_{h \in [H]}$ be a per-episode trajectory sample that consists of the trajectory for the learner and the adversary, and $\mathbb{P}_\theta^{\pi,\mu}$ be the trajectory distribution induced by the world model $\theta$, the learner's policy $\pi$ and the adversary's policy $\mu$. The learner's episodic value is defined as

$$V_\theta^{\pi,\mu} := \mathbb{E}_{\tau \sim \mathbb{P}_\theta^{\pi,\mu}} \Big[ \sum_{h=1}^{H} r_{A,h}(o_{A,h}) \Big],$$

i.e., the total expected reward the learner accumulates over $H$ steps under the world model $\theta$, when the learner follows policy $\pi$ while the adversary follows policy $\mu$.

**The second player as an adaptive adversary.** We consider the adaptive adversaries, following the framework of (Nguyen-Tang & Arora, 2024; 2025). In particular, an adaptive adversary is allowed to adapt to the learner's past strategies. That is, the adversary in episode $t$ is characterized by a deterministic response map

$$\mathcal{R}_t : \ \Pi^t \to \Psi, \qquad (\pi^1, \ldots, \pi^t) \mapsto \mu^t,$$

which depends on the entire learner policy history up to and including $\pi^t$. For a policy $\pi$, let $[\pi]^t := (\pi, \ldots, \pi)$ denote the $t$-fold repetition.

This adaptive response generalizes the canonical Stackelberg game, where the defender (the learner) commits a strategy and the follower (the adaptive adversary) selects her response strategy accordingly, to the setting where the adversary can remember all the learner's past strategies, not simply the learner's current-episode strategy as in Stackeberg games. That said, the adaptive adversary in our model is more general and powerful than the defender's response in Stackelberg games.

**Policy regret minimization.** We measure the learner's performance against adaptive adversaries using the notion of policy regret (Arora et al., 2012), which compares the learner's cumulative reward to that of the best fixed policy sequence in hindsight, accounting for the adaptive nature of the adversary. In particular, the learner's policy regret over a sequence of $T$ policies $\pi^1, \ldots, \pi^T$ is

$$\mathrm{PR}(T) \ := \ \sup_{\pi \in \Pi} \sum_{t=1}^{T} \Big( V_{\theta^*}^{\pi, \mathcal{R}_t([\pi]^t)} \ - \ V_{\theta^*}^{\pi^t, \mathcal{R}_t(\pi^1, \ldots, \pi^t)} \Big),$$

where $\mathcal{R}_t([\pi]^t)$ is the adversary's response under the counterfactual history in which the learner plays $\pi$ in episodes $1{:}t$, and $\theta^*$ is the groundtruth world model.

## 3 STRUCTURAL ASSUMPTIONS

Learning is intractable in general, without structural assumptions. In this section, we introduce natural assumptions on the adversary behavior and the POMG.

### 3.1 ADVERSARY BEHAVIOR MODEL

It is now well-established that learning in Markov games against adaptive adversaries who are memory-unbounded, non-stationary or unstructurally responsive is not sample-efficient (Nguyen-Tang & Arora, 2024; 2025). Since Markov games are a subclass of POMGs, the learning hardness for policy regret minimization extends from Markov games to POMGs. Thus, to ensure tractable learning, we impose the following assumptions on the behavior of the adaptive adversary, extending the similar assumptions by (Nguyen-Tang & Arora, 2024; 2025) for Markov games to POMGs.

**Assumption 1.** *For brevity, write the policy block $\pi^{u:v} := (\pi^u, \ldots, \pi^v)$ (with $u \leq v$) and set $\bar{t} := \max\{1, \ t - m + 1\}$. The adversary response functions $\{\mathcal{R}_t\}_{t \in \mathbb{N}}$ satisfy the following conditions:*

1. $m$-***memory bounded.*** *There exist $m \geq 0$ and a mapping $g_t : \Pi^m \to \Psi$ such that, for all $t$,*

$$\mathcal{R}_t(\pi^{1:t}) \ = \ g_t\big(\pi^{\bar{t}:t}\big).$$

2. ***Stationary.*** *The reaction rule is time-invariant: there is a fixed $g : \Pi^m \to \Psi$ with*

$$\mathcal{R}_t(\pi^{1:t}) \ = \ g\big(\pi^{\bar{t}:t}\big) \qquad \text{for all } t.$$

3. ***Posterior-Lipschitz.*** *Given a learner policy block $\pi^{1:m}$, let $P^{\pi^{1:m}, g, \theta}(\tau_A, \tau_B)$ denote the induced joint trajectory distribution. For any step $h$ and any adversary trajectory $\tau_B$, define the posterior-predictive policy*

$$S_{\tau_B}(\pi_h^i) \ := \ \mathbb{E}_{\tau_A \sim P^{\pi^{1:m}, g, \theta}(\cdot | \tau_B)} \big[ \pi_h^i(\cdot \mid \tau_A) \big] \in \mathbb{R}^A.$$

*where the expectation is with respect to the conditional law of $\tau_A$ given $\tau_B$ under the joint trajectory measure $P^{\pi^{1:m}, g, \theta}$.*

*Let $g(\cdot \mid \tau_B, \pi^{1:m})_h$ denote the distribution of the adversary's action at step $h$ induced by the response rule $g$ given private trajectory $\tau_B$ and learner policies $\pi^{1:m}$. Then there exists a constant $L \geq 0$ such that, for any two policy blocks $\pi^{1:m}, \nu^{1:m}$, any $h \in [H]$, and any adversary trajectory $\tau_B$,*

$$\big\| g(\cdot \mid \tau_B, \pi^{1:m})_h - g(\cdot \mid \tau_B, \nu^{1:m})_h \big\| \ \leq \ L \max_{i \in [m]} \big\| S_{\tau_B}(\pi_h^i) - S_{\tau_B}(\nu_h^i) \big\|.$$

While the bounded-memory and stationarity assumptions directly follow prior work on Markov games (Nguyen-Tang & Arora, 2024; 2025), our introduction of Posterior-Lipschitz is novel. This condition requires that if two policy blocks of the learner induce similar posterior action distributions given a fixed adversary trajectory, then the adversary's response distributions must also be similar. We define this condition using posterior predictives because, in partially observable Markov games (POMGs), policies depend on histories, whereas in standard Markov games (MGs) it suffices to consider Markov policies.

Finally, we parameterize the entire game using a joint model $\xi = (\theta, \Phi)$, where $\theta$ represents the world model parameters and $\Phi$ represents the parameters for the adversary channel $g$. We denote $\zeta^* = (\theta^*, \Phi^*)$ the groundtruth parameters and assume that the learner has access to $\Theta \ni \theta^*$, $\Psi \ni \Phi$.

**Remark 1.** *The norm $\|\cdot\|$ can be any norm on the corresponding finite-dimensional spaces. Since all such norms are equivalent, a different choice would merely rescale the constant L. In the subsequent proofs, we will adopt the $\ell_1$ norm.*

As a running example, we consider a linear response adversary model, motivated by the linear model considered initially in (Nguyen-Tang & Arora, 2025).

**Example 1** (Linear response Adversary). *There exist $d_{\mathrm{adv}} \in \mathbb{Z}_{>0}$, a nonnegative column-stochastic matrix $\Phi^\star \in \mathbb{R}_+^{B \times d_{\mathrm{adv}}}$, and weights $w_h^\pi(\tau_{B,h-1}) \in \mathbb{R}^{d_{\mathrm{adv}}}$ with $\|w_h^\pi(\tau_{B,h-1})\|_1 = \mathcal{O}(1)$ such that*

$$g_h(\cdot \mid \tau_{B,h-1}, \pi^{[m]}) = \Phi^\star w_h^\pi(\tau_{B,h-1}) \in \mathbb{R}^B \qquad \forall h.$$

We note that if the weights $w_h^\pi(\tau_{B,h-1})$ are L-Lipschitz with respect to $\pi$, then the adversary defined above is also L-Posterior-Lipschitz.

## 3.2 Multi-step Weakly Revealing POMGs

Learning POMGs is notoriously intractable in general. In this paper, we consider policy regret minimization in a rich class of POMGs that satisfy the weakly revealing conditions. Weakly revealing is a standard identifiability condition in the POMDP (Liu et al., 2022a) and POMG (Liu et al., 2022b) literature. Intuitively, over a length-$\kappa$ observation window, distinct latent states induce distinguishable distributions of observable sequences under fixed action prefixes.

Fix an enumeration of the learner's observation space $\mathcal{O}_A$ with $|\mathcal{O}_A| = O_A$. For a window length $\kappa \geq 1$, define the $\kappa$-step emission–action matrix $M_h^{(\kappa)} \in \mathbb{R}^{(O_A^\kappa A^{\kappa-1} B^{\kappa-1}) \times S}$ by

$$\left[M_h^{(\kappa)}\right]_{((a_{h:h+\kappa-2}, b_{h:h+\kappa-2}), o_{h:h+\kappa-1}), s} := \Pr(o_{h:h+\kappa-1} \mid s_h = s, a_{h:h+\kappa-2}, b_{h:h+\kappa-2}). \quad (1)$$

**Definition 1** (Multi-step $(\kappa, \alpha_\kappa)$-weakly revealing). *A POMG is $(\kappa, \alpha_\kappa)$-weakly revealing if*

$$\min_{h \in [H-\kappa+1]} \sigma_{\min}\left(M_h^{(\kappa)}\right) \geq \alpha_\kappa,$$

*where $\sigma_{\min}(\cdot)$ denotes the smallest singular value. This tacitly requires $O_A^\kappa A^{\kappa-1} B^{\kappa-1} \geq S$ so that $M_h^{(\kappa)}$ can have full column rank.*

**Remark 2** (Single-step weakly revealing). *For each $h \in [H]$, write the one-step emission kernel $E_h(\cdot \mid s)$ as a matrix $\mathbb{O}_h \in \mathbb{R}^{O_A \times S}$ with entries $[\mathbb{O}_h]_{o,s} = \Pr(o_h = o \mid s_h = s)$. When $\kappa = 1$, we have $M_h^{(1)} = \mathbb{O}_h$, so Definition 1 reduces to the single-step $\alpha$-weakly revealing condition $\min_{h \in [H]} \sigma_{\min}(\mathbb{O}_h) \geq \alpha_1$.*

## 4 Main Results

In this section, we present our algorithmic framework and theoretical analysis.

### 4.1 Unified Algorithmic Framework via Mini-batched Optimistic MLE

In this section, we introduce MOMLE, our novel model-based algorithm designed for Partially Observable Markov Games. The pseudocode is provided in Algorithm 1.

The core idea of MOMLE is to adapt and fundamentally redesign the high-level batched optimism framework, previously developed for fully-observable Markov games Nguyen-Tang & Arora (2024), to meet the unique challenges of partial observability. The shift from full observability to partial observability necessitates a move from a hybrid value-and-model-based approach to a purely joint model-learning strategy.

The high-level procedure of the algorithm follows a periodic pattern:

---

**Algorithm 1** MOMLE: Mini-batched Optimistic MLE

---

**Require:** Confidence parameters $\alpha, \beta_t$, number of batches $K$, adversary memory $m$, the learner's policy class $\Pi$, weakly revealing parameter $\kappa$.

1: **Initialize:**
2: World model confidence set $\mathcal{W} \leftarrow \{\theta \in \Theta : \sigma_{min}(M_h^{(\kappa)}) \geq \alpha\}$
3: Adversary model class $\Psi$ that parameterizes Assumption 1
4: Joint confidence set $\mathcal{C} \leftarrow \Xi := \mathcal{W} \times \Psi$
5: History dataset $\mathcal{D} \leftarrow \emptyset$
6: $\pi_{\text{current}} \leftarrow$ arbitrary initial policy
7: **for** batch $j = 1, \ldots, K$ **do**
8:     Select optimistic policy-model pair $(\pi_{\text{new}}, \xi_{\text{new}}) \in \arg\max_{(\pi, \xi) \in \Pi \times \mathcal{C}} V^\pi(\xi)$.
9:     **if** $\pi_{\text{new}} \neq \pi_{\text{current}}$ **then**
10:         Execute $\pi_{\text{new}}$ for $m - 1$ episodes (warm-up phase, discard data).
11:         $\pi_{\text{current}} \leftarrow \pi_{\text{new}}$.
12:     **end if**
13:     Execute $\pi_{\text{current}}$ for $\lfloor T/K \rfloor$ episodes.
14:     Add all collected learner trajectory pairs $(\pi_{\text{current}}, \tau_A)$ to $\mathcal{D}$.
15:     Update the joint confidence set $\mathcal{C}$ based on the joint log-likelihood over all data in $\mathcal{D}$:

$$\mathcal{C} \leftarrow \left\{ \xi \in \Xi : \sum_{(\pi^i, \tau_A^i) \in \mathcal{D}} \log \mathbb{P}_\xi^{\pi^i}(\tau_A^i) \geq \sup_{\xi' \in \Xi} \sum_{(\pi^i, \tau_A^i) \in \mathcal{D}} \log \mathbb{P}_{\xi'}^{\pi^i}(\tau_A^i) - \beta \right\}$$

16: **end for**

---

- **Optimistic planning (Line 8):** At the start of batch $j$, search the current joint confidence set $\mathcal{C}_{j-1}$ and pick a new optimistic policy–model pair that attains the largest predicted value.

- **Data collection with warm up(Line 9-Line 14):** For a newly selected policy, execute $m - 1$ warm-up episodes to stabilize the adversary's response and discard the warm-up data. Once stabilized, run the same policy for $\lfloor T/K \rfloor$ episodes and collect all learner trajectories.

- **Periodic updates (Line 15):** At the end of each batch, the algorithm updates the joint model confidence set using all historical data, based on the maximum likelihood principle.

**Joint Modeling via a Single Confidence Set.** A fundamental challenge in POMGs is that the learner only has access to its own trajectory, $\tau_A$, where the influence of the world dynamics ($\theta$) and the opponent's strategy ($\Phi$) are intrinsically entangled. This creates a severe identifiability challenge: from the learner's data alone, it is often impossible to uniquely attribute an observed outcome to either the world's stochasticity or the opponent's strategic choice. Consequently, attempting to learn two separate confidence sets for $\theta$ and $\Phi$ may fail to achieve sample-efficient learning. To resolve this, the MOMLE algorithm employs a cornerstone strategy: it maintains a single, joint confidence set $\mathcal{C} \subseteq \Xi$ over the entire system parameter space $\xi = (\theta, \Phi)$.

This design is crucial and provides three key advantages. First, it aligns the learning task with what is actually possible, by targeting the joint parameter $\xi$ that is statistically identifiable from the learner's data. Second, it enables a valid application of the optimism principle directly on the joint parameters that govern the true value $V^\pi(\xi)$. Third, by updating the confidence set based on the joint log-likelihood $\sum_{(\pi^i, \tau_A^i) \in \mathcal{D}} \log \mathbb{P}_\xi^{\pi^i}(\tau_A^i)$, we ensure that the set of plausible models reliably shrinks as more data is collected, leading to controlled regret. Here $\mathbb{P}_\xi^\pi(\tau)$ denotes the probability of observable history $\tau$ under model $\xi$ and policy $\pi$.

## 4.2 THEORETICAL ANALYSIS

Our regret analysis is built upon the framework of the Observable Operator Model (OOM). First proposed by (Jaeger, 2000), OOMs provide an alternative parameterization for partially observable systems that allows for a linear-algebraic treatment of their dynamics. This approach has recently been central to proving the tractability of single-agent POMDPs (Liu et al., 2022a). We adapt this framework to the Stackelberg POMG setting, where it enables a decomposition of the game's complex, coupled dynamics into tractable components.

### 4.2.1 A Reduction from any POMG to an Augmented POMDP

Consider the augmented hidden state $s'_h = (s_h, \zeta, \tau_{A,h-1})$ with fixed policy memory $\zeta = [\pi]^m$ and learner history $\tau_{A,h-1}$. On the learner marginal, the per-step observation kernel and controlled transition are defined by

$$\mathbb{O}'_h(o|s'_h) := \mathbb{O}^A_h(o|s_h), \qquad T'_h(s'_{h+1}|s'_h, a_h) := \mathbb{P}(s'_{h+1}|s'_h, a_h),$$

where $\mathbb{O}'_h(o|s'_h)$ lifts the orginal emission matrix to the augmented hidden state and $T'_h(\cdot|s'_h, a_h)$ is the transition law induced by the following generator: From the current world state $s_h$, the learner's observation $o_{A,h}$ is emitted. The adversary's action is marginalized via the learner-projected response and the world then transitions to $s_{h+1}$ under control $a_h$. Finally, the history $\tau_{A,h-1}$ and memory $\zeta$ in hidden state are updated deterministically.

Hence, the learner–observable process is a finite POMDP on $\mathcal{S}'$ with kernels $(\mathbb{O}'_h, T'_h)$.

**Remark 3.** *In the Stackelberg POMG, once the learner makes an action $a_h$, the adversary's response can be folded into the environment, i.e."adversary + world" becomes a single aggressive world, which is a partially observable process with augmented state $s'_h$, while the learner's information remains unchanged.*

### 4.2.2 Operator Decomposition

We extend the standard OOM results from finite POMDPs (Liu et al., 2022a) to our setting of Stackeberg OOM, where we can construct a operator representation of POMG.

**Lemma 1** (Stackelberg OOM Factorization Lemma). *Fix a learner policy $\pi$. With the above assumptions, the learner–observable process under any parameter $\xi$ admits a finite-dimensional controlled OOM representation. Specifically, there exist a nonnegative prediction state $q_0^\xi \in \mathbb{R}_+^{O_A}$ and nonnegative one-step operators $J_h^{\xi,\pi}(o_h, a_h) \in \mathbb{R}_+^{O_A \times O_A}$ satisfying:*

*1. (Factorization) For any marginal learner trajectory $\tau_A = (o_{A,1}, a_1, \ldots, o_{A,H}, a_H)$,*

$$\mathbb{P}_\xi^\pi(\tau_A) = \mathbf{1}^\top J_H^{\xi,\pi}(o_{A,H}, a_H) \cdots J_1^{\xi,\pi}(o_{A,1}, a_1) q_0^\xi, \qquad \mathbf{1} := (1, \ldots, 1)^\top.$$

*2. (Causal decomposition) Each one-step operator decomposes as*

$$J_h^{\xi,\pi}(o_h, a_h) = G_h^{\Phi,\pi}(o_h, a_h) W_h^\theta, \text{ where}$$

- $W_h^\theta \in \mathbb{R}_+^{O_A \times O_A}$, *called the **world channel** (the "leader" who acts first), is a nonnegative linear map, independent of $(o_{A,h}, a_h)$, that advances the predictive vector on $\mathbb{R}_+^{O_A}$ using only the world parameters $\theta$,*
- $G_h^{\Phi,\pi}(o_{A,h}, a_h) \in \mathbb{R}_+^{O_A \times O_A}$, *called the **adversary channel** (the "follower" who acts subsequently), is a family of nonnegative linear maps indexed by $(o_{A,h}, a_h)$, aggregating the learner emission and the follower's response under $(\Phi, \pi)$.*

*3. (Normalization) For all $b \in \mathbb{R}_+^{O_A}$, $\sum_{(o,a)} \mathbf{1}^\top J_h^{\xi,\pi}(o, a) q = \mathbf{1}^\top q$, so $\sum_{(o,a)} J_h^{\xi,\pi}(o, a)$ is stochastic on $\mathbb{R}_+^{O_A}$.*

*4. (Stability) There exists $c(\alpha) = \widetilde{\mathcal{O}}(1/\alpha)$ such that for all $h$ and all $v \in \mathbb{R}^{O_A}$,*

$$\mathbb{E}\left[\left\| J_H^{\xi,\pi}(O_H, A_H) \cdots J_{h+1}^{\xi,\pi}(O_{h+1}, A_{h+1}) v \right\|_1 \mid \tau_h\right] \leq c(\alpha_\kappa) \|v\|_1,$$

*Here, $\alpha_\kappa$ is the weakly-revealing parameter in Def. 1*

After reducing the Stackelberg POMG to a learner-marginal POMDP, the standard OOM yields a linear operator representation of the trajectory law. These operators also satisfy the standard Normalization and Stability properties, ensuring they collectively define a valid probability model.

**On the causal decomposition (vs. the standard result in (Liu et al., 2022a)).** While we built on the OOM result of (Liu et al., 2022a), our Stackelberg setting presents a unique challenge that the techniques used in (Liu et al., 2022a) do not suffice. In particular, each per-step operator $J_h^{\xi,\pi}$ is a *coupled* black box whose joint dependence on $\xi = (\theta, \Phi)$. Our core technical idea is a causal decomposition that opens this box, separating the world channel $W^\theta$ from the adversary channel $G^{\Phi,\pi}$. A detailed proof is given in Appendix C.1.

### 4.3 ELUDER CONDITIONS

Our analysis for batch update is built upon the Eluder condition, a structural complexity measure that generalizes the pigeonhole principle and the elliptical potential lemma, which are foundational for proving sample efficiency in MDPs (Jin et al., 2018) and POMDPs (Liu et al., 2023).

More specifically, the batched nature of our algorithm requires a slightly stronger variant known as the $\ell_2$-type Eluder condition (Xiong et al., 2023), and we refer the reader to Nguyen-Tang & Arora (2025, Sec. 5.1) for additional background, examples, and a comparison with the standard Eluder dimension.

**Definition 2** (Eluder dimensions for the $\kappa$-length world operator class and the adversary operator classes). *Let $\kappa \in \mathbb{N}$ that represents the window length of the history, we define two classes of scalar-valued functions that characterize the $\kappa$-length history-conditioned distribution errors:*

- *The world model operator class* $\mathcal{F}_{\Theta}^{[\kappa]} := \left\{ (\pi, \tau_{h-1}) \mapsto \sum_{t=h}^{h+\kappa-1} \left\| \left( W_t^{\theta} - W_t^{\theta^\star} \right) q_{t-1}^{\xi^\star} \right\|_1^2 : \theta \in \Theta \right\}$

- *The adversary model operator class* $\mathcal{G}_{\Psi}^{[\kappa]} := \left\{ (\pi, \tau_{h-1}, o, a) \mapsto \right.$
  $\sum_{t=h}^{h+\kappa-1} \mathbb{E}_{(o_t, a_t) \sim p^\star(\cdot|\tau_{t-1}; \pi)} \left\| \left( G_t^{\Phi, \pi}(o_t, a_t) - G_t^{\Phi^\star, \pi}(o_t, a_t) \right) q_t^{\mathrm{mid},\star} \right\|_1^2 : \Phi \in \Psi \right\}$

*where $G_h^{\Phi, \pi}$ and $W_h^{\theta}$ are the causal decomposition of $J_h^{\xi, \pi}$, i.e., $J_h^{\xi, \pi}(o, a) = G_h^{\Phi, \pi}(o, a) W_h^{\theta}$ (see Lemma 1), and*

$$q_{h-1}^{\xi^\star}(\pi, \tau_{h-1}) := \frac{J_{h-1}^{\xi^\star, \pi}(o_{h-1}, a_{h-1}) \cdots J_1^{\xi^\star, \pi}(o_1, a_1) b_0^{\xi^\star}}{\mathbf{1}^\top \left( J_{h-1}^{\xi^\star, \pi}(o_{h-1}, a_{h-1}) \cdots J_1^{\xi^\star, \pi}(o_1, a_1) b_0^{\xi^\star} \right)}, \qquad q_h^{\mathrm{mid},\star} := W_h^{\theta^\star} b_{h-1}^{\xi^\star}.$$

*Let $dim_E(\mathcal{F}_{\Theta}^{[\kappa]})$ and $dim_E(\mathcal{G}_{\Psi}^{[\kappa]})$ be the Eluder dimension (Definition 3) of $\mathcal{F}_{\Theta}^{[\kappa]}$ and $\mathcal{G}_{\Psi}^{[\kappa]}$.*

**Definition 3** ($\ell_2$-type Eluder dimension). *Let $F$ be a class of nonnegative scalar functions. We say $F$ has $\ell_2$-type Eluder dimension $\dim_E(F) = d$ if $d$ is the smallest integer such that for any input sequence $x^{1:T}$, any model sequence $\{f^i\}_{i=1}^T \subset F$, and any $\lambda > 0$, the following holds:*

$$\text{if } \forall t \in [T] : \sum_{i=1}^{t-1} f^t(x^i) \leq \lambda, \quad \text{then } \sum_{i=1}^{t} f^i(x^i) \leq C \, d \, \lambda \log t,$$

*where $C > 0$ is a universal constant.*

Intuitively, if functions in $F$ fit the past data well on average, then large squared errors can occur on at most $O(d \log T)$ rounds, so the total prediction error can be controlled by the Eluder dimension.

Similar to Example 1, we consider a linear world model as a running example to concretize the discussions on our Eluder dimensions.

**Example 2** (**Linear World Model**). *There exist a nonnegative column-stochastic matrix $W^\star \in \mathbb{R}_+^{O_A^\kappa \times d_w}$, and weights $u_h^\pi(\tau_{A,h-1}) \in \mathbb{R}^{d_w}$ with $\|u_h^\pi(\tau_{A,h-1})\|_1 \leq 1$ such that*

$$q_h^{\mathrm{mid},\star}(\pi, \tau_{A,h-1}) = W^\star u_h^\pi(\tau_{A,h-1}) \in \mathbb{R}^{O_A^\kappa} \qquad \forall h.$$

**Lemma 2** (Eluder dimension for linear operator classes). *Under Examples 1 and 2, the world and adversary operator classes satisfy: $\dim_E(\mathcal{F}_{\Theta}^{[\kappa]})) = \widetilde{\mathcal{O}}(d_w O_A^\kappa)$, and $\dim_E(\mathcal{G}_{\Psi}^{[\kappa]}) = \widetilde{\mathcal{O}}(d_{\mathrm{adv}} B)$,*

*Proof see Appendix F.2.*

### 4.4 POLICY REGRET BOUNDS

We now present the main theoretical contribution of this paper. The following theorem provides an upper bound on the policy regret for the MOMLE algorithm (Algorithm 1) operating under the key assumptions detailed in the problem setup. The theorem establishes that as long as the learning problem satisfies the single step $\alpha$-weakly revealing and the adversary is $m$-memory, stationary, and posterior-Lipschitz, our algorithm can achieve sublinear policy regret.

**Covering number.** Let $(\mathcal{X}, d)$ be a pseudometric space. For any $\varepsilon > 0$, an $\varepsilon$-cover is a finite subset $\mathcal{X}_\varepsilon \subseteq \mathcal{X}$ such that $\sup_{x \in \mathcal{X}} \inf_{x' \in \mathcal{X}_\varepsilon} d(x, x') \leq \varepsilon$. The $\varepsilon$-covering number is $N(\varepsilon; \mathcal{X}, d) := \min\{|\mathcal{X}_\varepsilon| : \mathcal{X}_\varepsilon \text{ is an } \varepsilon\text{-cover of } (\mathcal{X}, d)\}$.

We are now ready to state our main theorem.

**Theorem 1** (Policy Regret Bound for MOMLE). *Fix any $\delta \in (0, 1)$. Set the joint confidence radius in Algorithm 1 as follows:*

$$\beta = c\big(\log N(1/T; \Xi, d_\Xi) + \log(K/\delta)\big), \text{ where } d_\Xi(\xi, \xi') := \sup_{\pi \in \Pi} \left\| \mathbb{P}_\xi^\pi - \mathbb{P}_{\xi'}^\pi \right\|_1$$

*for an absolute constant $c > 0$. With probability at least $1 - \delta$, choosing $K = \left\lceil \sqrt{d_{E,[\kappa]}\, T} \right\rceil$ batches in Algorithm 1 yields a total policy regret*

$$PR(T) = \widetilde{\mathcal{O}}\Big( H\, (m + \sqrt{\beta})\, \sqrt{d_{E,[\kappa]}\, T} \Big),$$

*where $d_{E,[\kappa]} := \dim_E(\mathcal{F}_\Theta^{[\kappa]}) + \dim_E(\mathcal{G}_\Psi^{[\kappa]})$ is the total Eluder dimension of the world and adversary classes.*

**Corollary 1** (**Instantiation for linear world & adversary**). *Under the linear model in Examples 1 and 2, choosing the $K = \left\lceil \sqrt{d_{E,[\kappa]}\, T} \right\rceil$ batches in Algorithm 1 yields a total policy regret:*
$PR(T) = \widetilde{\mathcal{O}}\Big( H\, (m + \sqrt{\beta})\, \sqrt{(d_w\, O_A^\kappa + d_{\mathrm{adv}}\, B)\, T} \Big).$

**Comparison with prior work.** Specializing our POMG framework to the fully observable Markov game setting yields the regret bound $\mathrm{PR}(T) = \widetilde{\mathcal{O}}\Big( H\, (m + \sqrt{\beta})\, \sqrt{d_E T} \Big)$. The bound preserves its mathematical structure, while the Eluder dimension $d_E$ simplifies to reflect the less complex environment. We compare this result to the bound for the BOVL algorithm presented in Nguyen-Tang & Arora (2025), which reports the policy–regret bound $\mathrm{PR}(T) = \mathcal{O}\Big( \bar{V}\, (H + m)\sqrt{d_E\, \gamma\, T\, \log^3 T} \Big)$. The apparent difference in our results stems from two accounting choices: we normalize the value scale such that $\bar{V} = 1$ and include the $(m-1)K$ warm-up episodes within the total time horizon $T$. If we adopt the same conventions as Nguyen-Tang & Arora (2025) by retaining the scale $\bar{V}$ and excluding the warm-up period with an effective horizon of $T_{\mathrm{eff}} = T - (m-1)K$, our bound reduces to the same order as theirs.

**Proof overview of Theorem 1.** Our proof consists of the main four steps.

1. **Optimism in Joint Confidence Sets (Appendix A and B)** In each batch $j$, we maintain a joint Maximum Likelihood Estimation (MLE) confidence set $\mathcal{C}_j$ for the world and adversary models. On the high-probability event that the true model parameters $\xi^\star$ are within $\mathcal{C}_j$ for all batches, our optimistic policy selection reduces the per-batch regret to a value difference. This difference is further bounded by the Total Variation distance between the process distributions induced by the optimistic model $\hat{\xi}_j$ and the true model $\xi^\star$.

2. **Regret Decomposition via Causal Telescoping (Appendix C)** We represent the learner-observable process using OOMs, which permit a causal factorization of the one-step transition operator into a world operator ($G_h$) and an adversary operator ($W_h$). A novel telescoping sum decomposition then breaks down the TV distance into a sum of horizon-step errors stemming from the world model estimation and the adversary model estimation.

3. **From Likelihood Bounds to Quadratic Constraints (Appendix D)** We translate the statistical log-likelihood bound that defines the confidence set $\mathcal{C}_j$ into a powerful analytical tool. Leveraging the weakly-revealing property of the environment, this bound is converted into a set of crucial quadratic constraints on the $\ell_1$-norms of the operator errors.

4. **Bounding Regret via a Batched Eluder Argument (Appendix E)** Finally, we bound the cumulative regret by summing the decomposed errors. We define a batch as "bad" if its operator estimation error is large. The quadratic constraints ensure that a "bad" batch is highly informative. An $\ell_2$-Eluder dimension argument then bounds the total number of possible "bad" batches. This, combined with a simple bound for "good" batches, yields the final $\widetilde{\mathcal{O}}(\sqrt{T})$ policy regret.

## 5 CONCLUSION AND DISCUSSION

In this work, we develop the first algorithmic framework and theoretical analysis for policy regret minimization in multi-step weakly revealing partially observable Markov games. We establish the first $\mathcal{O}(\sqrt{T})$ policy regret bound through a novel analysis framework that builds upon a joint maximum likelihood estimation (MLE) algorithm and a decoupling argument based on the causal decomposition of world and adversary models. Future research directions include extending our framework to incorporate function approximation and expanding the class of learnable partially observable environments for policy regret minimization.

ETHICS STATEMENT

This paper presents a purely theoretical study of multi-agent reinforcement learning. Our work does not involve any datasets, human subjects, or the deployment of physical systems. As such, there are no direct ethical issues concerning data privacy, algorithmic bias, or immediate societal harm.

REPRODUCIBILITY STATEMENT

This paper is of a purely theoretical nature. To ensure the reproducibility of our results, we have provided detailed and self-contained proofs for all theorems, propositions, and lemmas in the appendix. We believe the provided proofs are sufficient for an expert in the field to verify the correctness of our claims.

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

CONTENTS

## LLM USAGE STATEMENT

We utilized a large language model to assist with the writing and polishing of this manuscript. Its role was strictly limited to improving the linguistic quality of the text by refining language, enhancing readability, and ensuring clarity. All scientific contributions, including the core ideas, research methodology, and proofs, were developed exclusively by the authors.

## A VALIDITY OF CONFIDENCE SETS

We use the joint parameter $\xi = (\theta, \Phi)$ and define, for each batch $j$,

$$L_j(\xi) := \sum_{(\pi^i, \tau_A^i) \in \mathcal{D}_j} \log \mathbb{P}_\xi^{\pi^i}(\tau_A^i), \qquad \mathcal{C}_j(\beta) := \{\xi \in \Xi : L_j(\xi) \geq \sup_{\xi' \in \Xi} L_j(\xi') - \beta\}.$$

Planning at the start of batch $j+1$ uses the projections

$$\mathcal{W}_{j+1} := \{\theta : \exists \Phi \text{ s.t. } (\theta, \Phi) \in \mathcal{C}_j(\beta)\}, \qquad \Psi_{j+1} := \{\Phi : \exists \theta \text{ s.t. } (\theta, \Phi) \in \mathcal{C}_j(\beta)\}.$$

### A.1 OPTIMISTIC $\varepsilon$-NET AND MGF BOUND

Let $d(\xi, \xi') := \sup_\pi \|\mathbb{P}_\xi^\pi - \mathbb{P}_{\xi'}^\pi\|_1$ be the TV metric over learner-observable trajectory laws and let $N(\varepsilon; \Xi, d)$ be the covering number. We take an optimistic $\varepsilon$-net $\bar{\Xi} \subset \Xi$ with $\varepsilon = T^{-1}$ and set

$$\beta = c\Big(\log N(T^{-1}, \Xi, d) + \log(K/\delta)\Big),$$

which is the confidence radius used in Theorem1.

**Lemma 3** (Joint MGF bound under optimistic discretization). *Let $\mathcal{D}_N = \{(\pi^i, \tau_A^i)\}_{i=1}^N$ be any (possibly adaptive) sequence of policies and observed learner trajectories. Fix any $\bar{\xi} \in \bar{\Xi}$ such that $\sup_\pi \|\mathbb{P}_{\bar{\xi}}^\pi - \mathbb{P}_{\xi^\star}^\pi\|_1 \leq T^{-1}$. Then*

$$\mathbb{E}\left[\exp\Big(\sum_{i=1}^N \log \frac{\mathbb{P}_{\bar{\xi}}^{\pi^i}(\tau_A^i)}{\mathbb{P}_{\xi^\star}^{\pi^i}(\tau_A^i)}\Big)\right] \leq e.$$

*Proof.* Identical in structure to Liu et al. (2022a, Prop. 13), replacing the single parameter $\theta$ with the joint parameter $\xi = (\theta, \Phi)$ and full trajectories by the learner-observable marginals $\tau_A$ (marginalization preserves normalization). Let $\mathcal{F}_i$ be the history up to episode $i-1$ and define $r_i := \mathbb{P}_{\bar{\xi}}^{\pi^i}(\tau_A^i)/\mathbb{P}_{\xi^\star}^{\pi^i}(\tau_A^i)$. By the tower property, $\mathbb{E}[r_i \mid \mathcal{F}_i] = \sum_{\tau_A} \mathbb{P}_{\bar{\xi}}^{\pi^i}(\tau_A) \leq 1 + T^{-1}$, hence $\mathbb{E}[\exp(\sum_{i=1}^N \log r_i)] = \mathbb{E}[\prod_{i=1}^N \mathbb{E}[r_i \mid \mathcal{F}_i]] \leq (1 + T^{-1})^N \leq e$. $\qquad \square$

### A.2 MARKOV + UNION BOUND AND BATCHED VALIDITY

**Proposition 1** (Validity of Batched Confidence Sets). *Let the confidence parameter $\beta$ be a constant defined as $\beta := c\big(\log|\bar{\Xi}| + \log(K/\delta)\big)$. With probability at least $1 - \delta$, the true parameter $\xi^\star$ is contained in the confidence set $\mathcal{C}_j(\beta)$ for all batches $j \in \{1, \ldots, K\}$.*

*Proof.* Let $N_j = |\mathcal{D}_j|$ be the number of episodes collected up to the end of batch $j$. We apply Lemma 3 to the dataset $\mathcal{D}_j$ for a fixed $\bar{\xi} \in \bar{\Xi}$ and a fixed batch $j \in \{1, \ldots, K\}$. By Markov's inequality,

$$\mathbb{P}\left(\sum_{(\pi^i, \tau_A^i) \in \mathcal{D}_j} \log \frac{\mathbb{P}_{\bar{\xi}}^{\pi^i}(\tau_A^i)}{\mathbb{P}_{\xi^\star}^{\pi^i}(\tau_A^i)} > \log(K|\bar{\Xi}|/\delta)\right) \leq \frac{\mathbb{E}[\exp(\ldots)]}{K|\bar{\Xi}|/\delta} \leq \frac{e \cdot \delta}{K|\bar{\Xi}|}.$$

Taking a union bound over all $j \in \{1, \ldots, K\}$ and all $\bar{\xi} \in \bar{\Xi}$, the probability that the bound is violated for any pair $(j, \bar{\xi})$ is at most $K \cdot |\bar{\Xi}| \cdot \frac{e \cdot \delta}{K|\bar{\Xi}|} = e \cdot \delta$. Rescaling $\delta$ appropriately, we have that with probability at least $1 - \delta$, for all $j \in \{1, \ldots, K\}$ and all $\bar{\xi} \in \bar{\Xi}$:

$$\sum_{(\pi^i, \tau_A^i) \in \mathcal{D}_j} \log \frac{\mathbb{P}_{\bar{\xi}}^{\pi^i}(\tau_A^i)}{\mathbb{P}_{\xi^\star}^{\pi^i}(\tau_A^i)} \le c(\log |\bar{\Xi}| + \log(K/\delta)).$$

By the optimistic property of the discretization ($\mathbb{P}_\xi \le \mathbb{P}_{\bar{\xi}}$), the bound also holds for all $\xi \in \Xi$. The proposition's claim then follows directly from the definition of $\mathcal{C}_j(\beta)$. $\qquad\square$

**Corollary 2** (Validity for Planning on the Joint Confidence Set). *Under the high-probability event of Proposition 1, we have*

$$\xi^\star \in \mathcal{C}_j(\beta) \qquad \textit{for all } j \in \{1, \ldots, K\}.$$

*Proof.* Hence any planning step at the beginning of batch $j+1$ that optimizes an objective over the joint confidence set $\mathcal{C}_j(\beta)$ is valid in the sense that the true parameter $\xi^\star$ is feasible. $\qquad\square$

## B    POLICY REGRET AND OPTIMISM

We fix the true joint parameter $\xi^\star = (\theta^\star, \Phi^\star)$. Let $\mathcal{R}_t(\pi^1, \ldots, \pi^t)$ denote the realized adversary in round $t$, and let $\mathcal{R}_t([\pi]^t)$ denote the counterfactual response had the learner played the comparator policy $\pi$ for all the first $t$ episodes. Per-step rewards lie in $[0, 1]$ and the horizon is $H$.

### B.1    OPTIMISM REPLACEMENT IN THE BATCHED FRAMEWORK

At the beginning of each batch $j \in \{1, \ldots, K\}$, the algorithm selects a fixed optimistic pair $(\pi_j, \xi_j)$ from the confidence set $\mathcal{C}_{j-1}(\beta)$:

$$(\pi_j, \xi_j) \in \arg \max_{\pi \in \Pi, \, \xi \in \mathcal{C}_{j-1}(\beta)} V^{\pi, \, \mathcal{R}(\pi)}(\xi),$$

and keeps $(\pi_t, \xi_t) = (\pi_j, \xi_j)$ for all $t \in \text{Batch}_j$.

**Lemma 4** (Optimism per Batch). *On the high-probability event that $\xi^\star \in \mathcal{C}_{j-1}(\beta)$ for all batches $j \in \{1, \ldots, K\}$, it holds for every batch $j$ and every comparator policy $\pi$ that, for all data-collection rounds $t \in \text{Batch}_j$,*

$$V^{\pi, \, \mathcal{R}_t([\pi]^t)}(\xi^\star) \; \le \; V^{\pi_j, \, \mathcal{R}(\pi_j)}(\xi_j).$$

*Proof.* Fix $j$ and $\pi$. Consider any $t \in \text{Batch}_j$ belonging to the data-collection part of batch $j$ (i.e., after the $(m-1)$-episode warm-up under $\pi_j$). Along the counterfactual path $[\pi]^t$, the last $m$ policy blocks are $\pi$. Along the counterfactual path $[\pi]^t$, the last $m$ policy blocks are all equal to $\pi$, so by stationarity and $m$-memory the opponent's response depends only on this repeated block and we have $\mathcal{R}_t([\pi]^t) = \mathcal{R}(\pi)$. Therefore $V^{\pi, \, \mathcal{R}_t([\pi]^t)}(\xi^\star) = V^{\pi, \, \mathcal{R}(\pi)}(\xi^\star)$. Since $\xi^\star \in \mathcal{C}_{j-1}(\beta)$ and $(\pi_j, \xi_j)$ maximizes $V^{\pi, \, \mathcal{R}(\pi)}(\xi)$ over $\Pi \times \mathcal{C}_{j-1}(\beta)$, we have $V^{\pi, \, \mathcal{R}(\pi)}(\xi^\star) \le V^{\pi_j, \, \mathcal{R}(\pi_j)}(\xi_j)$. This holds for every such $t$. $\qquad\square$

Consequently, for $t \in \text{Batch}_j$ the executed policy is $\pi_j$ and the realized adversary is $\mathcal{R}_t(\pi^1, \ldots, \pi^t)$; thus

$$\text{PR}(T) \le \sum_{j=1}^K \sum_{t \in \text{Batch}_j} \left( V^{\pi_j, \, \mathcal{R}(\pi_j)}(\xi_j) - V^{\pi_j, \, \mathcal{R}_t(\pi^1, \ldots, \pi^t)}(\xi^\star) \right). \qquad (2)$$

**Remark 4** (Within-batch stationarity). *Each batch $j$ begins with an $(m-1)$-episode warm-up under $\pi_j$. By stationarity, $m$-memory, and SLC, the opponent stabilizes to the fixed response $\mathcal{R}_{\xi^\star}(\pi_j)$ on all data-collection rounds of batch $j$. Let $n_j$ denote the number of data-collection episodes in batch $j$ (excluding warm-up). Then*

$$PR(T) \le \sum_{j=1}^K n_j \left( V^{\pi_j, \mathcal{R}_{\xi^\star}(\pi_j)}(\xi_j) - V^{\pi_j, \mathcal{R}_{\xi^\star}(\pi_j)}(\xi^\star) \right).$$

where $n_j := |\mathrm{Batch}_j| = \lfloor T/K \rfloor$ denotes the number of data-collection episodes in batch $j$. If warm-up rounds are included in regret, they add at most $H(m-1)K$.

### B.2 SAME-POLICY VALUE–DISTRIBUTION BOUND

**Lemma 5.** *Assume per-episode returns satisfy $0 \le R(\tau_H) \le H$. For any fixed policy $\pi$, any two joint models $\xi_1, \xi_2$, and their corresponding opponent responses $\mathcal{R}_1, \mathcal{R}_2$,*

$$V^{\pi,\mathcal{R}_1}(\xi_1) \;-\; V^{\pi,\mathcal{R}_2}(\xi_2) \;\le\; H \left\| \mathbb{P}^{\pi,\mathcal{R}_1}_{\xi_1} \;-\; \mathbb{P}^{\pi,\mathcal{R}_2}_{\xi_2} \right\|_1.$$

*Proof.* Write $V^{\pi,\mathcal{R}}(\xi) = \sum_{\tau_H} \mathbb{P}^{\pi,\mathcal{R}}_\xi(\tau_H)\, R(\tau_H)$ with $R(\tau_H) \in [0, H]$. Let $\Delta(\tau_H) = \mathbb{P}^{\pi,\mathcal{R}_1}_{\xi_1}(\tau_H) - \mathbb{P}^{\pi,\mathcal{R}_2}_{\xi_2}(\tau_H)$. Then

$$V^{\pi,\mathcal{R}_1}(\xi_1) - V^{\pi,\mathcal{R}_2}(\xi_2) = \sum_{\tau_H} \Delta(\tau_H)\, R(\tau_H) \;\le\; \sum_{\tau_H} |\Delta(\tau_H)|\, \|R\|_\infty$$

$$\le H \sum_{\tau_H} |\Delta(\tau_H)| \;=\; H \left\| \mathbb{P}^{\pi,\mathcal{R}_1}_{\xi_1} - \mathbb{P}^{\pi,\mathcal{R}_2}_{\xi_2} \right\|_1.$$

$\square$

Applying Lemma 5 to the inner term of equation 2 and using Remark 4 (i.e., $\mathcal{R}_t = \mathcal{R}_{\xi^\star}(\pi_j)$ on data-collection rounds) gives

$$PR(T) \;\le\; H \sum_{j=1}^{K} n_j \cdot \left\| \mathbb{P}^{\pi_j, \mathcal{R}_{\xi_j}(\pi_j)}_{\xi_j} - \mathbb{P}^{\pi_j, \mathcal{R}_{\xi^\star}(\pi_j)}_{\xi^\star} \right\|_1. \tag{3}$$

## C  POMG TELESCOPING VIA OPERATOR DECOMPOSITION

Fix a learner policy $\pi$ and work with marginal learner–trajectory prefixes $\tau_h = (o_1, a_1, \ldots, o_h, a_h)$. Within the *data-collection* rounds (i.e., after the $(m-1)$-episode warm-up), the response faced by a fixed $\pi$ is time-invariant by Remark 4. Thus the learner-observable process under $\pi$ is time-homogeneous on these rounds.

### C.1  CONTROLLED OOM REPRESENTABILITY AND STABILITY

*Proof of Lemma 1.* The proof proceeds by establishing the equivalence of the POMG with a finite-state POMDP, then leveraging this equivalence to derive the existence, causal decomposition, and stability of its OOM/PSR representation.

**Proof of Part 1 (Factorization)**   Consider the augmented hidden state $s'_h = (s_h, \zeta, \tau_{A,h-1})$, where $s_h \in \mathcal{S}$ is the world state, $\zeta = [\pi]^m \in \mathcal{Z}_{\mathrm{pol}}$ is the fixed policy memory within the batch, and $\tau_{A,h-1} \in (\mathcal{O}_A \times \mathcal{A})^{h-1}$ is the learner-side within-episode history. Within a batch, the learner uses a fixed policy $\pi$ and the adversary is stationary and posterior-Lipschitz. At step $h$:

1. (*Adversary response*) $b_h \sim \mu^\star(\cdot \mid \zeta, \tau_{A,h-1})$.

2. (*Observation*) $o_h \sim \mathbb{O}^A_h(\cdot \mid s_h)$ (lift of the original emission to the augmented state).

3. (*World transition*) $s_{h+1} \sim T_h(\cdot \mid s_h, a_h, b_h)$.

4. (*Memory update*) $\tau_{A,h} = (\tau_{A,h-1}, o_h, a_h)$ and $\zeta' = \zeta$ deterministically.

Define the *joint* one-step kernel of the learner observation and the next augmented state:

$$\mathbb{P}(o_h, s'_{h+1} \mid s'_h, a_h) = \sum_{b_h \in \mathcal{B}} \mu^\star(b_h \mid \zeta, \tau_{A,h-1})\, \mathbb{O}^A_h(o_h \mid s_h)\, T_h(s_{h+1} \mid s_h, a_h, b_h)$$

$$\times\; \mathbf{1}\{\zeta' = \zeta,\; \tau_{A,h} = (\tau_{A,h-1}, o_h, a_h)\}. \tag{4}$$

On the learner marginal, the per-step observation kernel and the controlled transition are the marginals of equation 4:

$$O'_h(o_h \mid s'_h) := \mathbb{O}_h^A(o_h \mid s_h) = \sum_{s'_{h+1}} \mathbb{P}(o_h, s'_{h+1} \mid s'_h, a_h), \tag{5}$$

$$T'_h(s'_{h+1} \mid s'_h, a_h) := \mathbb{P}(s'_{h+1} \mid s'_h, a_h) = \sum_{o \in \mathcal{O}_A} \mathbb{P}(o, s'_{h+1} \mid s'_h, a_h). \tag{6}$$

With observation-based rewards $r_{A,h} : \mathcal{O}_A \to [0,1]$,

$$R'_h(s'_h, a_h) = \sum_{o \in \mathcal{O}_A} \mathbb{O}_h^A(o \mid s_h)\, r_{A,h}(o). \tag{7}$$

Therefore, within a batch the learner–observable process is a finite POMDP on

$$\mathcal{S}' = \mathcal{S} \times \mathcal{Z}_{\mathrm{pol}} \times \bigcup_{k=0}^{H-1} (\mathcal{O}_A \times \mathcal{A})^k,$$

and the joint law of $(o_h, s'_{h+1})$ depends only on $(s'_h, a_h)$. By standard OOM results for finite POMDPs in (Liu et al., 2022a), there exist a dimension $d$, an initial vector $q_0^\xi \in \mathbb{R}_+^d$, and nonnegative one-step operators $J_h^{\xi,\pi}(o_h, a_h)$ such that, for any learner trajectory $\tau_A = (o_1, a_1, \ldots, o_H, a_H)$,

$$\mathbb{P}_\xi^\pi(\tau_A) = \mathbf{1}^\top J_H^{\xi,\pi}(o_H, a_H) \cdots J_1^{\xi,\pi}(o_1, a_1)\, q_0^\xi. \tag{8}$$

**Proof of Part 2 (Causal Decomposition)** Work on the augmented space of Part 1 with the step-$h$ distribution $\eta_h$ over $s'_h = (s_h, \zeta, \tau_{h-1})$.

**(i) World channel $\widehat{W}_h^\theta$.** Given $\eta_h$ and learner action $a_h$, define a nonnegative kernel that propagates the world state while *indexing* by a hypothetical opponent action $b_h$:

$$(\widehat{W}_h^\theta \eta_h)(s'_h, s_{h+1}, b_h; a_h) := \eta_h(s'_h)\, T_h(s_{h+1} \mid s_h, a_h, b_h).$$

This map depends only on the world kernel $T_h$ and carries forward $(s_h, \zeta, \tau_{h-1})$ for downstream use.

**(ii) Adversary channel $\widehat{G}_h^{\Phi,\pi}(o_h, a_h)$.** Acting on $m_h := \widehat{W}_h^\theta \eta_h$, it marginalizes $b_h$ using the adversary response and emits the learner-side observation, while deterministically updating the history:

$$(\widehat{G}_h^{\Phi,\pi}(o_h, a_h) m_h)(o_h, s'_{h+1})$$
$$:= \sum_{s'_h} \sum_{b_h} \mu^\Phi(b_h \mid \zeta, \tau_{h-1})\, \mathbb{O}_h^A(o_h \mid s_h)\, \mathbf{1}\{\zeta' = \zeta,\ \tau_h = (\tau_{h-1}, o_h, a_h)\}\, m_h(s'_h, s_{h+1}, b_h; a_h), \tag{9}$$

where $s'_{h+1} = (s_{h+1}, \zeta', \tau_h)$. This map carries all dependence on $(\Phi, \pi)$ through $\mu^\Phi(\cdot \mid \zeta, \tau_{h-1})$ and $(\zeta, \tau_{h-1})$ embedded in $s'_h$.

Define the hidden-layer one-step joint kernel as the composition

$$\widehat{K}_h^{\xi,\pi}(o_h, a_h) := \widehat{G}_h^{\Phi,\pi}(o_h, a_h)\, \widehat{W}_h^\theta,$$

so that for any $s'_h$ it yields $\mathbb{P}_\xi^\pi(o_h, s'_{h+1} \mid s'_h, a_h)$ (cf. equation 4 with $\mu^*$ and $\mathbb{O}_h^A$).

**Transport to the predictive-state space.** By finite-rank realization, there exist parameter-independent linear maps $\mathcal{L}_h : \mathbb{R}^d \to \mathbb{R}^{|\mathcal{S}'|}$ and $\mathcal{P}_h : \mathbb{R}^{|\mathcal{S}'|} \to \mathbb{R}^d$ such that

$$J_h^{\xi,\pi}(o_h, a_h) = \mathcal{P}_h\, \widehat{K}_h^{\xi,\pi}(o_h, a_h)\, \mathcal{L}_h.$$

Insert an identity factorization $I = \mathcal{Q}_h \mathcal{R}_h$ on the hidden space with $\mathcal{Q}_h, \mathcal{R}_h$ linear and parameter-independent, and set

$$W_h^\theta := \mathcal{R}_h\, \widehat{W}_h^\theta\, \mathcal{L}_h, \qquad G_h^{\Phi,\pi}(o_h, a_h) := \mathcal{P}_h\, \widehat{G}_h^{\Phi,\pi}(o_h, a_h)\, \mathcal{Q}_h.$$

Then

$$J_h^{\xi,\pi}(o_h, a_h) = G_h^{\Phi,\pi}(o_h, a_h)\, W_h^\theta. \tag{10}$$

which is the desired causal factorization.

**Proof of Part 3 (Normalization)**    Because the operators arise from conditional probability kernels of a finite controlled POMDP, they are nonnegative and mass-preserving. Concretely, (Liu et al., 2022a) shows that probabilities of historys and next-observations can be written as operator products (their Eq. (36)), which for any fixed action $a$ implies

$$\sum_{o \in \mathcal{O}_A} \mathbf{1}^\top J_h^{\xi,\pi}(o,a)\, q \;=\; \mathbf{1}^\top q \qquad \text{for all } q \in \mathbb{R}_+^d.$$

Equivalently, $\sum_o J_h^{\xi,\pi}(o,a)$ is stochastic on $\mathbb{R}_+^d$ for every $a$. If the learner randomizes actions according to $\pi(\cdot \mid \tau_{h-1})$, then

$$\sum_{a \in \mathcal{A}} \pi(a \mid \tau_{h-1}) \sum_{o \in \mathcal{O}_A} \mathbf{1}^\top J_h^{\xi,\pi}(o,a)\, q \;=\; \mathbf{1}^\top q.$$

**Proof of Part 4 (Stability)**    Under the $\kappa$-step $\alpha_\kappa$-weakly revealing assumption, the block OOM telescoping argument of Liu et al. (2022a, Appx. F.1 and Lemma 31) applies verbatim to $J_h^{\xi,\pi} = G_h^{\Phi,\pi} W_h^\theta$. Hence there exists $C(\alpha_\kappa) = \widetilde{\mathcal{O}}(\mathrm{poly}(1/\alpha_\kappa))$ such that, for any $h \in \{1, \ldots, H-1\}$, any $v \in \mathbb{R}^d$, and any prefix $\tau_h$,

$$\mathbb{E}\Big[\big\| J_H^{\xi,\pi}(O_H, A_H) \cdots J_{h+1}^{\xi,\pi}(O_{h+1}, A_{h+1})\, v \big\|_1 \,\Big|\, \tau_h\Big] \;\leq\; C(\alpha_\kappa)\, \|v\|_1. \qquad \square$$

### C.2   Two-stage telescoping bound

We derive a two-stage telescoping bound that separates, at each step, the world and adversary contributions to the same-policy distributional gap. Let

$$T_H^{\xi,\pi}(\tau_A) := J_H^{\xi,\pi}(o_H, a_H) \cdots J_1^{\xi,\pi}(o_1, a_1).$$

Define unnormalized predictive states

$$q_{h-1}^{\xi^\star}(\tau_{h-1}) := J_{h-1}^{\xi^\star,\pi}(o_{h-1}, a_{h-1}) \cdots J_1^{\xi^\star,\pi}(o_1, a_1)\, q_0^{\xi^\star}, \qquad q_h^{\mathrm{mid},\xi^\star}(\tau_{h-1}) := W_h^{\theta^\star}\, q_{h-1}^{\xi^\star}(\tau_{h-1}).$$

**Lemma 6** ($\kappa$-step two-stage telescoping under weakly revealing). *Fix a policy $\pi$ and two joint models $\xi = (\theta, \Phi)$ and $\xi^\star = (\theta^\star, \Phi^\star)$. Assume per-step factorization $J_t^{\xi,\pi} = G_t^{\Phi,\pi} W_t^\theta$ with normalization (all maps are nonnegative and $\ell_1$-nonexpansive after summing over emitted symbols), and assume the model is $\kappa$-step $\alpha_\kappa$-weakly revealing so that the $\kappa$-step controlled tail is $\ell_1$-stable with constant $C(\alpha_\kappa) = \widetilde{\mathcal{O}}(\mathrm{poly}(1/\alpha_\kappa))$. Partition the horizon into consecutive blocks $I_r = \{h_r, \ldots, \min(h_r + \kappa - 1, H)\}$ with $h_r = (r-1)\kappa + 1$. Let $q_{t-1}^{\xi^\star}$ be the normalized predictive state under $(\xi^\star, \pi)$ and $q_t^{\mathrm{mid},\xi^\star} := W_t^{\theta^\star} q_{t-1}^{\xi^\star}$. Then the total-variation distance between trajectory laws satisfies*

$$\begin{aligned}
\big\| \mathbb{P}_\xi^\pi - \mathbb{P}_{\xi^\star}^\pi \big\|_1 &\leq \big\| q_0^\xi - q_0^{\xi^\star} \big\|_1 \\
&\quad + C(\alpha_\kappa) \sum_r \sum_{t \in I_r} \Big\{ \big\| (W_t^\theta - W_t^{\theta^\star})\, q_{t-1}^{\xi^\star} \big\|_1 \\
&\quad + \mathbb{E}_{(o_t, a_t) \sim p_{\xi^\star}^\pi(\cdot | \tau_{t-1})} \big\| (G_t^{\Phi,\pi} - G_t^{\Phi^\star,\pi})(o_t, a_t)\, q_t^{\mathrm{mid},\xi^\star} \big\|_1 \Big\}.
\end{aligned}$$

*Proof.* For integers $u \leq v$ write $J_{u:v}^\xi := J_v^\xi \cdots J_u^\xi$ and $J_{u:u-1}^\xi := I$. For each block $I_r$, the product-difference identity gives

$$J_{h_r:h_r+\kappa-1}^\xi - J_{h_r:h_r+\kappa-1}^{\xi^\star} = \sum_{t \in I_r} \left( J_{t+1:h_r+\kappa-1}^\xi \right) \left( J_t^\xi - J_t^{\xi^\star} \right) \left( J_{h_r:t-1}^{\xi^\star} \right). \qquad (11)$$

Using the per-step split

$$J_t^\xi - J_t^{\xi^\star} = \underbrace{(G_t^{\Phi,\pi} - G_t^{\Phi^\star,\pi})\, W_t^{\theta^\star}}_{\text{adversary}} + \underbrace{G_t^{\Phi,\pi}\, (W_t^\theta - W_t^{\theta^\star})}_{\text{world}}, \qquad (12)$$

and applying the above to the nonnegative state $q_{h_r-1}^{\xi^\star}$, we obtain by the triangle inequality

$$\left\| \left( J_{h_r:h_r+\kappa-1}^{\xi} - J_{h_r:h_r+\kappa-1}^{\xi^\star} \right) q_{h_r-1}^{\xi^\star} \right\|_1 \leq \sum_{t \in I_r} \left( T_{t,r}^{\mathrm{adv}} + T_{t,r}^{\mathrm{world}} \right), \tag{13}$$

where

$$T_{t,r}^{\mathrm{adv}} := \left\| J_{t+1:h_r+\kappa-1}^{\xi} \left( G_t^{\Phi,\pi} - G_t^{\Phi^\star,\pi} \right) W_t^{\theta^\star} \times J_{h_r:t-1}^{\xi^\star} q_{h_r-1}^{\xi^\star} \right\|_1,$$

$$T_{t,r}^{\mathrm{world}} := \left\| J_{t+1:h_r+\kappa-1}^{\xi} G_t^{\Phi,\pi} \left( W_t^\theta - W_t^{\theta^\star} \right) \times J_{h_r:t-1}^{\xi^\star} q_{h_r-1}^{\xi^\star} \right\|_1.$$

Normalization implies $\sum_{(o_t,a_t)} \left\| G_t^{\Phi,\pi}(o_t,a_t) x \right\|_1 \leq \|x\|_1$ for all $x \geq 0$ (and similarly for $G_t^{\Phi^\star,\pi}$). By $\kappa$-step weakly revealing, there is $C(\alpha_\kappa)$ such that for any $v \geq 0$,

$$\mathbb{E}\left[ \left\| J_{t+1:h_r+\kappa-1}^{\xi^\star} v \right\|_1 \,\Big|\, \tau_t \right] \leq C(\alpha_\kappa) \|v\|_1, \qquad \left\| J_{h_r:t-1}^{\xi^\star} v \right\|_1 \leq \|v\|_1. \tag{14}$$

For the world term, set $x := J_{h_r:t-1}^{\xi^\star} b_{h_r-1}^{\xi^\star} = q_{t-1}^{\xi^\star}$. Then

$$\mathbb{E}\left[ T_{t,r}^{\mathrm{world}} \right] \leq C(\alpha_\kappa) \left\| G_t^{\Phi,\pi}(W_t^\theta - W_t^{\theta^\star}) x \right\|_1 \leq C(\alpha_\kappa) \left\| (W_t^\theta - W_t^{\theta^\star}) q_{t-1}^{\xi^\star} \right\|_1. \tag{15}$$

For the adversary term, with $q_t^{\mathrm{mid},\xi^\star} := W_t^{\theta^\star} x$,

$$\mathbb{E}\left[ T_{t,r}^{\mathrm{adv}} \right] \leq C(\alpha_\kappa) \left\| (G_t^{\Phi,\pi} - G_t^{\Phi^\star,\pi}) q_t^{\mathrm{mid},\xi^\star} \right\|_1 \leq C(\alpha_\kappa) \sum_{(o_t,a_t)} \left\| (G_t^{\Phi,\pi} - G_t^{\Phi^\star,\pi})(o_t,a_t) q_t^{\mathrm{mid},\xi^\star} \right\|_1. \tag{16}$$

Moreover, $\kappa$-step weakly revealing implies a lower bound on the conditional mass over supported $(o_t, a_t)$, hence

$$\sum_{(o_t,a_t)} \left\| (G_t^{\Phi,\pi} - G_t^{\Phi^\star,\pi})(o_t,a_t) q_t^{\mathrm{mid},\xi^\star} \right\|_1$$
$$\leq C(\alpha_\kappa) \, \mathbb{E}_{(o_t,a_t) \sim p_{\xi^\star}^\pi(\cdot|\tau_{t-1})} \left\| (G_t^{\Phi,\pi} - G_t^{\Phi^\star,\pi})(o_t,a_t) q_t^{\mathrm{mid},\xi^\star} \right\|_1, \tag{17}$$

absorbing this factor into $C(\alpha_\kappa)$.

Finally, taking expectations in the block bound, summing over $t \in I_r$ and over all $r$, and adding the initial-state discrepancy yields the claimed inequality. $\qquad\square$

### C.3 FROM SIGNATURES TO OPERATORS

**Lemma 7** (Lipschitz Transfer). *Assume Posterior-Lipschitz and the factorization $J_h^{\xi,\pi} = G_h^{\Phi,\pi} W_h^\theta$ with normalization (Lemma 1). Then there exists $L_G = \mathcal{O}(L)$ such that for any $h$, policies $\pi, \nu$, and $v \in \mathbb{R}_+^d$,*

$$\sum_{(o,a)} \left\| (G_h^{\Phi,\pi} - G_h^{\Phi,\nu})(o,a) v \right\|_1 \leq L_G \, \Delta_\sigma(\pi,\nu) \|v\|_1, \qquad \Delta_\sigma(\pi,\nu) := \max_{i \in [m]} \left\| S_{\tau_B}^i(\pi) - S_{\tau_B}^i(\nu) \right\|_1.$$

*The same bound holds with $\Phi$ replaced by $\Phi^\star$.*

*Proof.* By the causal factorization equation 10 and the adversary channel equation 9, together with the finite-rank realization in Sec. C.1, there exist nonnegative linear maps $\widetilde{R}_h(o,a,b) : \mathbb{R}_+^d \to \mathbb{R}_+^d$ (independent of $(\Phi,\pi)$) such that equation 18 holds.

$$G_h^{\Phi,\pi}(o,a) v = \sum_b g_h(b \mid \tau_B; \pi) \widetilde{R}_h(o,a,b) v \qquad (\forall \, v \in \mathbb{R}_+^d). \tag{18}$$

Since $\sum_{(o,a)} J_h^{\xi,\pi}(o,a)$ is stochastic for every $\pi$, taking $g_h$ as a point mass gives

$$\sum_{(o,a)} \widetilde{R}_h(o,a,b) \text{ is stochastic on } \mathbb{R}_+^d \Rightarrow \sum_{(o,a)} \left\| \widetilde{R}_h(o,a,b) v \right\|_1 \leq \|v\|_1 \quad (\forall \, v \in \mathbb{R}_+^d). \tag{19}$$

By equation 18,

$$\big(G_h^{\Phi,\pi} - G_h^{\Phi,\nu}\big)(o,a)\,v \;=\; \sum_b \Big[ g_h(b \mid \tau_B; \pi) - g_h(b \mid \tau_B; \nu) \Big]\, \widetilde{R}_h(o,a,b)\, v.$$

Summing over $(o,a)$ and using equation 19,

$$\sum_{(o,a)} \big\| \big(G_h^{\Phi,\pi} - G_h^{\Phi,\nu}\big)(o,a)\,v \big\|_1 \;\leq\; \Big( \sum_b \big| g_h(b \mid \tau_B; \pi) - g_h(b \mid \tau_B; \nu) \big| \Big) \|v\|_1.$$

By Posterior-Lipschitz, $\sum_b |g_h(b \mid \tau_B; \pi) - g_h(b \mid \tau_B; \nu)| \leq L\,\Delta_\sigma(\pi,\nu)$, which proves the claim with $L_G := L$. The case $\Phi^\star$ is identical. $\qquad\square$

# D   Constraints for Operator Estimates from Batched OMLE

This section converts the high-probability joint-likelihood guarantee (Proposition 1) into quantitative constraints on per-step operator errors.

Fix an arbitrary batch $j \in \{1, \dots, K\}$. Work on the high-probability event where the optimistic model $\xi_j = (\theta_j, \Phi_j)$ chosen for batch $j$ satisfies $\xi_j \in \mathcal{C}_{j-1}(\beta)$. Hence, for the historical dataset $\mathcal{D}_{j-1}$,

$$\sum_{(\pi^i, \tau_A^i) \in \mathcal{D}_{j-1}} \log \frac{\mathbb{P}_{\xi^\star}^{\pi^i}(\tau_A^i)}{\mathbb{P}_{\xi_j}^{\pi^i}(\tau_A^i)} \;\leq\; \beta.$$

For any episode $i$ and step $h$, let $p_\xi(\cdot \mid \tau_{h-1}; \pi^i) \in \Delta(\mathcal{O}_A \times \mathcal{A})$ be the one-step conditional. Let $q_{h-1}^{\xi^\star}(\tau_{h-1})$ be the *normalized* true prediction state and $q_h^{\mathrm{mid}, \xi^\star}(\tau_{h-1}) := W_h^{\theta^\star} q_{h-1}^{\xi^\star}(\tau_{h-1})$.

**Proposition 2** (Likelihood-to-Squared-TV Bound on Past Data). *At the beginning of batch $j$, for $\xi_j \in \mathcal{C}_{j-1}(\beta)$,*

$$\sum_{(\pi^i, \tau_A^i) \in \mathcal{D}_{j-1}} \sum_{h=1}^H \mathbb{E}_{\tau_{h-1} \sim \mathbb{P}_{\xi^\star}^{\pi^i}} \mathrm{KL}\big( p_{\xi^\star}(\cdot \mid \tau_{h-1}; \pi^i) \,\|\, p_{\xi_j}(\cdot \mid \tau_{h-1}; \pi^i) \big) \;\leq\; \beta, \qquad (20)$$

$$\sum_{(\pi^i, \tau_A^i) \in \mathcal{D}_{j-1}} \sum_{h=1}^H \mathbb{E}_{\tau_{h-1} \sim \mathbb{P}_{\xi^\star}^{\pi^i}} \big\| p_{\xi^\star}(\cdot \mid \tau_{h-1}; \pi^i) - p_{\xi_j}(\cdot \mid \tau_{h-1}; \pi^i) \big\|_1^2 \;\leq\; 2\,\beta. \qquad (21)$$

*Proof.* Taking expectation of the joint log-likelihood ratio under $\mathbb{P}_{\xi^\star}^{\pi^i}$ and using the chain rule,

$$\mathbb{E}_{\mathbb{P}_{\xi^\star}^{\pi^i}} \left[ \log \frac{\mathbb{P}_{\xi^\star}^{\pi^i}(\tau_A)}{\mathbb{P}_{\xi_j}^{\pi^i}(\tau_A)} \right] = \sum_{h=1}^H \mathbb{E}_{\tau_{h-1} \sim \mathbb{P}_{\xi^\star}^{\pi^i}} \mathrm{KL}\big( p_{\xi^\star}(\cdot \mid \tau_{h-1}; \pi^i) \,\|\, p_{\xi_j}(\cdot \mid \tau_{h-1}; \pi^i) \big).$$

Summing over $(\pi^i, \tau_A^i) \in \mathcal{D}_{j-1}$ gives equation 20. Pinsker's inequality, applied conditionally on each $\tau_{h-1}$, yields $\| p_{\xi^\star} - p_{\xi_j} \|_1^2 \leq 2\,\mathrm{KL}(p_{\xi^\star} \| p_{\xi_j})$, which implies equation 21 after summing and taking expectations. $\qquad\square$

**Corollary 3** (Cross-signature propagation of adversary errors). *For any step $h$, policies $\pi, \nu$, and $v \in \mathbb{R}_+^d$,*

$$\sum_{(o,a)} \big\| \big(G_h^{\Phi,\pi} - G_h^{\Phi^\star,\pi}\big)(o,a)\,v \big\|_1 \;\leq\; \sum_{(o,a)} \big\| \big(G_h^{\Phi,\nu} - G_h^{\Phi^\star,\nu}\big)(o,a)\,v \big\|_1 \;+\; 2 L_G\, \Delta_\sigma(\pi,\nu)\, \|v\|_1,$$

*where $\Delta_\sigma(\pi,\nu) := \max_{i \in [m]} \| S_{\tau_B}^i(\pi) - S_{\tau_B}^i(\nu) \|_1$ and $L_G$ is from Lemma 7.*

*Proof.* Triangle inequality: $\| G_h^{\Phi,\pi} - G_h^{\Phi^\star,\pi} \| \leq \| G_h^{\Phi,\pi} - G_h^{\Phi,\nu} \| + \| G_h^{\Phi,\nu} - G_h^{\Phi^\star,\nu} \| + \| G_h^{\Phi^\star,\nu} - G_h^{\Phi^\star,\pi} \|$. Apply Lemma 7 to the first and third terms. $\qquad\square$

**Lemma 8** (Conditional distribution Lipschitzness). *For any prefix $\tau_{h-1}$ and policy $\pi$,*

$$\left\| p_{\xi^\star}(\cdot \mid \tau_{h-1}; \pi) - p_\xi(\cdot \mid \tau_{h-1}; \pi) \right\|_1 \leq \left\| [J_h^{\xi,\pi} - J_h^{\xi^\star,\pi}] q_{h-1}^{\xi^\star} \right\|_1.$$

*Proof.* Let $q := q_{h-1}^{\xi^\star}$ with $\mathbf{1}^\top q = 1$. By Lemma **??**(1,3), $p_\xi(\cdot \mid \tau_{h-1}; \pi) = \mathbf{1}^\top J_h^{\xi,\pi}(\cdot) q$. Then

$$\|p_{\xi^\star} - p_\xi\|_1 = \sum_{(o,a)} \left| \mathbf{1}^\top (J_h^{\xi^\star,\pi} - J_h^{\xi,\pi})(o,a)\, q \right| \leq \sum_{(o,a)} \left\| (J_h^{\xi,\pi} - J_h^{\xi^\star,\pi})(o,a)\, q \right\|_1 = \|[J_h^{\xi,\pi} - J_h^{\xi^\star,\pi}]\, q\|_1.$$

$\square$

**Lemma 9** (One-step causal split). *For any prefix $\tau_{h-1}$, policy $\pi$, and $q := q_{h-1}^{\xi^\star} \geq 0$,*

$$\sum_{(o_h,a_h)} \left\| [J_h^{\xi,\pi}(o_h,a_h) - J_h^{\xi^\star,\pi}(o_h,a_h)]\, q \right\|_1 \leq \underbrace{\left\| [W_h^\theta - W_h^{\theta^\star}]\, q \right\|_1}_{\text{world}}$$

$$+ C_{\mathrm{cm}}(\alpha_\kappa)\, \mathbb{E}_{(o_h,a_h) \sim p_{\xi^\star}(\cdot|\tau_{h-1};\pi)} \underbrace{\left\| [G_h^{\Phi,\pi}(o_h,a_h) - G_h^{\Phi^\star,\pi}(o_h,a_h)]\, q_h^{\mathrm{mid},\xi^\star} \right\|_1}_{\text{adversary}},$$

*where $C_{\mathrm{cm}}(\alpha_\kappa) = \widetilde{\mathcal{O}}(\mathrm{poly}(1/\alpha_\kappa))$ depends only on the $\kappa$-step weakly revealing condition.*

*Proof.* Since $J_h^{\xi,\pi} = G_h^{\Phi,\pi} W_h^\theta$,

$$J_h^{\xi,\pi} - J_h^{\xi^\star,\pi} = G_h^{\Phi,\pi}(W_h^\theta - W_h^{\theta^\star}) + (G_h^{\Phi,\pi} - G_h^{\Phi^\star,\pi})W_h^{\theta^\star}.$$

Summing $\ell_1$-norms and using nonnegativity plus $\sum_{(o,a)} J_h^{\xi,\pi}(o,a)$ stochastic (Lemma 1(3)),

$$\sum_{(o,a)} \|G_h^{\Phi,\pi}(o,a)\, (W_h^\theta - W_h^{\theta^\star})q\|_1 \leq \|(W_h^\theta - W_h^{\theta^\star})q\|_1.$$

For the adversary term, for nonnegative $f$ and full-support $q'$, $\sum_{(o,a)} f(o,a) \leq (\max_{(o,a)} 1/q'(o,a))\, \mathbb{E}_{q'}[f(o,a)]$. Take $q' = p_{\xi^\star}(\cdot \mid \tau_{h-1}; \pi)$ and $f(o,a) = \|(G_h^{\Phi,\pi} - G_h^{\Phi^\star,\pi})(o,a)\, q_h^{\mathrm{mid},\xi^\star}\|_1$. The $\kappa$-step weakly revealing condition yields the controlled-mass bound $\max_{(o,a)} 1/q'(o,a) \leq C_{\mathrm{cm}}(\alpha_\kappa)$. $\square$

**Proposition 3** (Operator Quadratic Constraints on Past Data). *There exists $C(\alpha_\kappa) = \widetilde{\mathcal{O}}(\mathrm{poly}(1/\alpha_\kappa))$ such that, at the beginning of any batch $j$ and for $\xi_j = (\theta_j, \Phi_j) \in \mathcal{C}_{j-1}(\beta)$,*

$$\sum_{(\pi^i,\tau_A^i)\in\mathcal{D}_{j-1}} \sum_{h=1}^H \mathbb{E}_{\tau_{h-1}\sim\mathbb{P}_{\xi^\star}^{\pi^i}} \left\| [W_h^{\theta_j} - W_h^{\theta^\star}]\, q_{h-1}^{\xi^\star} \right\|_1^2 \leq C(\alpha_\kappa)\,\beta, \tag{22}$$

$$\sum_{(\pi^i,\tau_A^i)\in\mathcal{D}_{j-1}} \sum_{h=1}^H \mathbb{E}_{\substack{\tau_{h-1}\sim\mathbb{P}_{\xi^\star}^{\pi^i}\\(o_h,a_h)\sim p_{\xi^\star}(\cdot|\tau_{h-1};\pi^i)}} \left\| [G_h^{\Phi_j,\pi^i}(o_h,a_h) - G_h^{\Phi^\star,\pi^i}(o_h,a_h)]\, q_h^{\mathrm{mid},\xi^\star} \right\|_1^2 \leq C(\alpha_\kappa)\,\beta, \tag{23}$$

*and an analogous bound holds for the initial prediction state.*

*Proof.* From Proposition 2,

$$\sum_{(\pi^i,\tau_A^i)\in\mathcal{D}_{j-1}} \sum_{h=1}^H \mathbb{E}_{\tau_{h-1}\sim\mathbb{P}_{\xi^\star}^{\pi^i}} \left\| p_{\xi^\star}(\cdot \mid \tau_{h-1}; \pi^i) - p_{\xi_j}(\cdot \mid \tau_{h-1}; \pi^i) \right\|_1^2 \leq 2\beta. \tag{24}$$

Fix $(i,h)$. Let $q := q_{h-1}^{\xi^\star}$ and $q_h^{\mathrm{mid}} := W_h^{\theta^\star} q$.

*World channel.* The $\kappa$-step weakly revealing assumption implies that the block emission map $M_h^{(\kappa)}(\pi)$ admits a right inverse with $\|M_h^{(\kappa)\,\dagger}\|_{1\to1} \leq \mathrm{poly}(1/\alpha_\kappa)$ on the cone of reachable predictive states, while controlled OOM guarantees $\|M_h^{(\kappa)}\|_{1\to1} \leq 1$. Together these yield two-sided

$\ell_1$ bounds between conditional-distribution errors and operator perturbations, i.e., a bi-Lipschitz relation on reachable states. Hence there exists $\bar{C}_\theta(\alpha_\kappa) = \widetilde{\mathcal{O}}(\text{poly}(1/\alpha_\kappa))$ with

$$\left\| [W_h^{\theta_j} - W_h^{\theta^\star}] \, q \right\|_1 \leq \bar{C}_\theta(\alpha_\kappa) \left\| p_{\xi^\star} - p_{\xi_j} \right\|_1. \tag{25}$$

Squaring equation 25, taking expectation over $\tau_{h-1} \sim \mathbb{P}_{\xi^\star}^{\pi^i}$, summing over $(i, h)$, and invoking equation 24 gives equation 22 with constant $2 \bar{C}_\theta(\alpha_\kappa)^2$.

*Adversary channel.* Similarly, there exists $\widetilde{C}_\Phi(\alpha_\kappa) = \widetilde{\mathcal{O}}(\text{poly}(1/\alpha_\kappa))$ such that for all $(o_h, a_h)$ in the support of $p_{\xi^\star}(\cdot \mid \tau_{h-1}; \pi^i)$,

$$\left\| [G_h^{\Phi_j, \pi^i}(o_h, a_h) - G_h^{\Phi^\star, \pi^i}(o_h, a_h)] \, q_h^{\text{mid}} \right\|_1 \leq \widetilde{C}_\Phi(\alpha_\kappa) \left\| p_{\xi^\star} - p_{\xi_j} \right\|_1. \tag{26}$$

Squaring equation 26 and taking expectation over $(o_h, a_h) \sim p_{\xi^\star}(\cdot \mid \tau_{h-1}; \pi^i)$ yields

$$\mathbb{E}_{(o_h, a_h)} \left\| [G_h^{\Phi_j, \pi^i} - G_h^{\Phi^\star, \pi^i}] \, q_h^{\text{mid}} \right\|_1^2 \leq \widetilde{C}_\Phi(\alpha_\kappa)^2 \left\| p_{\xi^\star} - p_{\xi_j} \right\|_1^2.$$

Taking expectation over $\tau_{h-1}$, summing $(i, h)$, and using equation 24 gives equation 23 with constant $2 \widetilde{C}_\Phi(\alpha_\kappa)^2$.

Finally set $C(\alpha_\kappa) := 2 \max\{\bar{C}_\theta(\alpha_\kappa)^2, \widetilde{C}_\Phi(\alpha_\kappa)^2\}$. $\qquad\qquad\qquad\qquad\qquad\qquad\square$

# E    BOUNDING CUMULATIVE REGRET VIA BATCHED ELUDER ARGUMENT

We adapt the batched "estimation-to-regret" bridge used by Nguyen-Tang & Arora (2024): operator quadratic constraints obtained from past data (Proposition 3) are transported to the current batch and then converted into a linear-in-$K$ bound via an $\ell_2$-Eluder counting argument. The key method is a "bad batch" analysis ensuring that large in-batch errors occur only $\widetilde{\mathcal{O}}(\text{Eluder dim})$ many times.

A batch $j$ is bad if the optimistic model $\xi_j = (\theta_j, \Phi_j)$ has large in-batch squared error (on the true distribution) under the fixed policy $\pi_j$ of that batch. Define

$$\mathcal{E}_{\text{world}}(j) := \sum_{t \in \text{Batch}_j} \sum_{h=1}^{H} \mathbb{E}_{\tau_{h-1} \sim \mathbb{P}_{\xi^\star}^{\pi_j}} \left\| [W_h^{\theta_j} - W_h^{\theta^\star}] \, q_{h-1}^{\xi^\star} \right\|_1^2,$$

$$\mathcal{E}_{\text{adv}}(j) := \sum_{t \in \text{Batch}_j} \sum_{h=1}^{H} \mathbb{E}_{\substack{\tau_{h-1} \sim \mathbb{P}_{\xi^\star}^{\pi_j} \\ (o_h, a_h) \sim p_{\xi^\star}(\cdot \mid \tau_{h-1}; \pi_j)}} \left\| [G_h^{\Phi_j, \pi_j} - G_h^{\Phi^\star, \pi_j}] \, q_h^{\text{mid}, \xi^\star} \right\|_1^2.$$

Let $C(\alpha_\kappa), \beta$ be from Proposition 3. Define

$$\mathcal{K}_{\text{world}}^{\text{bad}} := \{j : \mathcal{E}_{\text{world}}(j) > C(\alpha_\kappa)\beta\}, \mathcal{K}_{\text{adv}}^{\text{bad}} := \{j : \mathcal{E}_{\text{adv}}(j) > C(\alpha_\kappa)\beta\}, \mathcal{K}^{\text{bad}} := \mathcal{K}_{\text{world}}^{\text{bad}} \cup \mathcal{K}_{\text{adv}}^{\text{bad}}.$$

## E.1    TRANSPORTING HISTORICAL OPERATOR CONSTRAINTS TO THE CURRENT BATCH

Fix batch $j$ and write $\pi := \pi_j$. Let $\mathcal{V}_{j-1}$ be the set of policies appearing in $\mathcal{D}_{j-1}$, and choose a nearest historical policy

$$\nu_j \in \arg \min_{\nu \in \mathcal{V}_{j-1}} \Delta_\sigma(\pi, \nu).$$

**Lemma 10** (Historical to Current Operator Control at Batch $j$). *There exist* $C_*(\alpha_\kappa), C'_*(\alpha_\kappa) = \widetilde{\mathcal{O}}(\text{poly}(1/\alpha_\kappa))$ *such that*

$$\sum_{k=1}^{j-1} \sum_{h=1}^{H} \mathbb{E}_{\tau_{h-1} \sim \mathbb{P}_{\xi^\star}^{\pi_k}} \left\| [W_h^{\theta_j} - W_h^{\theta^\star}] \, q_{h-1}^{\xi^\star} \right\|_1^2 \leq C_*(\alpha_\kappa) \, \beta, \tag{27}$$

$$\sum_{k=1}^{j-1} \sum_{h=1}^{H} \mathbb{E}_{\substack{\tau_{h-1} \sim \mathbb{P}_{\xi^\star}^{\pi_k} \\ (o_h, a_h) \sim p_{\xi^\star}(\cdot \mid \tau_{h-1}; \pi)}} \left\| [G_h^{\Phi_j, \pi}(o_h, a_h) - G_h^{\Phi^\star, \pi}(o_h, a_h)] \, q_h^{\text{mid}, \xi^\star} \right\|_1^2 \leq C_*(\alpha_\kappa) \, \beta$$

$$+ C'_*(\alpha_\kappa) \, \Delta_\sigma(\pi, \nu_j)^2 \, \Gamma_{j-1}. \tag{28}$$

*where* $\Gamma_{j-1} := \sum_{k=1}^{j-1} \sum_{h=1}^{H} \mathbb{E}_{\tau_{h-1} \sim \mathbb{P}_{\xi^\star}^{\pi_k}} \|q_h^{\text{mid}, \xi^\star}\|_1^2$ *is finite.*

*Proof.* World part equation 27 is Proposition 3 applied to $(\theta_j, \cdot)$, so $C_*(\alpha_\kappa) = C(\alpha_\kappa)$. For the adversary part, Corollary 3 with $v = q_h^{\mathrm{mid}, \xi^\star}$ gives

$$\sum_{(o,a)} \|[G_h^{\Phi_j, \pi} - G_h^{\Phi^\star, \pi}](o, a)\, v\|_1 \leq \sum_{(o,a)} \|[G_h^{\Phi_j, \nu_j} - G_h^{\Phi^\star, \nu_j}](o, a)\, v\|_1 + 2L_G\, \Delta_\sigma(\pi, \nu_j)\, \|v\|_1.$$

Taking $\mathbb{E}_{(o,a) \sim p_{\xi^\star}(\cdot | \tau_{h-1}; \pi)}$, squaring and using $(a + b)^2 \leq 2a^2 + 2b^2$ yields

$$\mathbb{E}\|[G_h^{\Phi_j, \pi} - G_h^{\Phi^\star, \pi}]\, v\|_1^2 \leq 2\, \mathbb{E}\|[G_h^{\Phi_j, \nu_j} - G_h^{\Phi^\star, \nu_j}]\, v\|_1^2 + 8L_G^2\, \Delta_\sigma(\pi, \nu_j)^2\, \|v\|_1^2.$$

Summing over $k < j, h \leq H$ and invoking Proposition 3 at $\nu_j$ proves equation 28 with $C'_*(\alpha_\kappa) = 8L_G^2$ (absorbing polynomial factors into $\widetilde{\mathcal{O}}(\mathrm{poly}(1/\alpha_\kappa))$). $\qquad\square$

### E.2 BOUNDING THE NUMBER OF BAD BATCHES VIA ELUDER DIMENSION

**Proposition 4** (Cardinality of Bad Batches). *Let $\mathcal{F}_\Theta^{[\kappa]}$ and $\mathcal{F}_\Psi^{[\kappa]}$ be the $\kappa$-window world/adversary error classes with $\ell_2$-Eluder dimensions $d_{E,[\kappa]}^{(\theta)}$ and $d_{E,[\kappa]}^{(\Phi)}$, respectively. On the high-probability event of Proposition 1,*

$$\left|\mathcal{K}_{\mathrm{world}}^{\mathrm{bad}}\right| \leq \widetilde{\mathcal{O}}(d_{E,[\kappa]}^{(\theta)}), \qquad \left|\mathcal{K}_{\mathrm{adv}}^{\mathrm{bad}}\right| \leq \widetilde{\mathcal{O}}(d_{E,[\kappa]}^{(\Phi)}).$$

*Proof of Proposition 4.* We give the full proof for $\left|\mathcal{K}_{\mathrm{world}}^{\mathrm{bad}}\right|$; the adversary case is analogous after transporting historical constraints to the current batch via Lemma 10.

**Classes and per-batch error.** Define the $\kappa$-window world error class

$$\mathcal{F}_\Theta^{[\kappa]} := \Big\{ (\pi, \tau_{h-1}) \mapsto \sum_{t=h}^{\min(h+\kappa-1, H)} \big\|[W_t^\theta - W_t^{\theta^\star}]\, q_{t-1}^{\xi^\star}\big\|_1^2 \; : \; \theta \in \Theta \Big\},$$

with $\ell_1$-Eluder dimension $d_{E,[\kappa]}^{(\theta)}$. For batch $j$, define

$$\mathcal{E}_{\mathrm{world}}^{[\kappa]}(j) := \sum_{h=1}^H \mathbb{E}_{\tau_{h-1} \sim \mathbb{P}_{\xi^\star}^{\pi_j}} \left[ \sum_{t=h}^{\min(h+\kappa-1, H)} \big\|[W_t^{\theta_j} - W_t^{\theta^\star}]\, q_{t-1}^{\xi^\star}\big\|_1^2 \right].$$

By definition of bad batches, $j \in \mathcal{K}_{\mathrm{world}}^{\mathrm{bad}}$ iff $\mathcal{E}_{\mathrm{world}}^{[\kappa]}(j) > C_0$, where we set $C_0 := C_*(\alpha_\kappa)\beta$ from Proposition 3 (absorbing universal constants).

**Step 1: Dyadic decomposition and per-level counting.** For each integer $i \geq 1$, let

$$\mathcal{K}_i := \Big\{ j \in \mathcal{K}_{\mathrm{world}}^{\mathrm{bad}} : \mathcal{E}_{\mathrm{world}}^{[\kappa]}(j) \in [\, C_0 2^{i-1},\, C_0 2^i) \Big\}.$$

Write $\mathcal{K}_i = \{j_1 < \cdots < j_M\}$ with $M = |\mathcal{K}_i|$. Then

$$\sum_{m=1}^M \mathcal{E}_{\mathrm{world}}^{[\kappa]}(j_m) \geq M\, C_0\, 2^{i-1}. \tag{29}$$

**Step 2: Historical precondition for each selected model.** Each $\theta_{j_m}$ is selected using only the historical data $\mathcal{D}_{j_m - 1}$. By Proposition 3 (world part),

$$\sum_{k < j_m} \sum_{t=1}^H \mathbb{E}_{\tau_{t-1} \sim \mathbb{P}_{\xi^\star}^{\pi_k}} \big\|[W_t^{\theta_{j_m}} - W_t^{\theta^\star}]\, q_{t-1}^{\xi^\star}\big\|_1^2 \leq C_0.$$

Since each $\kappa$-window $\sum_{t=h}^{\min(h+\kappa-1, H)}(\cdot)$ overlaps any fixed step $t$ at most $\kappa$ times, the same historical budget controls the $\kappa$-window loss up to a factor $\kappa$:

$$\sum_{k < j_m} \sum_{h=1}^H \mathbb{E}_{\tau_{h-1} \sim \mathbb{P}_{\xi^\star}^{\pi_k}} \left[ \sum_{t=h}^{\min(h+\kappa-1, H)} \big\|[W_t^{\theta_{j_m}} - W_t^{\theta^\star}]\, q_{t-1}^{\xi^\star}\big\|_1^2 \right] \leq \kappa\, C_0, \tag{30}$$

which we absorb into $C_0$ henceforth.

Thus, for the sequence $\{\theta_{j_m}\}_{m=1}^M$, each element fits all *past* data with $\kappa$-window squared-loss budget $C_0$ by equation 30, yet incurs *new* loss at least $C_0 2^{i-1}$ on its own batch by equation 29. The $\ell_2$-type Eluder counting principle (applied to squared $\ell_1$ losses over $\mathcal{F}_\Theta^{[\kappa]}$ with dimension $d_{E,[\kappa]}^{(\theta)}$) gives

$$\sum_{m=1}^M \mathcal{E}_{\text{world}}^{[\kappa]}(j_m) \;\leq\; \widetilde{\mathcal{O}}\big(d_{E,[\kappa]}^{(\theta)} C_0 \, 2^{\,i}\big). \tag{31}$$

Comparing equation 29 and equation 31 yields $|\mathcal{K}_i| \;\leq\; \widetilde{\mathcal{O}}\big(d_{E,[\kappa]}^{(\theta)}\big)$.

**Step 3: Summation over levels.** Let $\mathrm{MaxError}$ denote the maximum feasible $\mathcal{E}_{\text{world}}^{[\kappa]}(j)$ (polynomially bounded). Then

$$\mathcal{K}_{\text{world}}^{\text{bad}} \subseteq \bigcup_{i=1}^{\lceil \log_2(\mathrm{MaxError}/C_0)\rceil} \mathcal{K}_i,$$

so

$$\big|\mathcal{K}_{\text{world}}^{\text{bad}}\big| \;\leq\; \sum_i |\mathcal{K}_i| \;\leq\; \widetilde{\mathcal{O}}\big(d_{E,[\kappa]}^{(\theta)}\big),$$

absorbing the logarithmic factor into $\widetilde{\mathcal{O}}(\cdot)$.

**Adversary case.** Transport historical constraints to the *current* batch's policy $\pi_j$ via Lemma 10:

$$\sum_{k<j}\sum_{h=1}^H \mathbb{E}_{\substack{\tau_{h-1}\sim\mathbb{P}_{\xi^\star}^{\pi_k} \\ (o_h,a_h)\sim p_{\xi^\star}(\cdot\,|\tau_{h-1};\pi_j)}} \left[\sum_{t=h}^{\min(h+\kappa-1,\,H)} \big\|[G_t^{\Phi_j,\pi_j}(o_t,a_t) - G_t^{\Phi^\star,\pi_j}(o_t,a_t)]\, q_t^{\mathrm{mid},\xi^\star}\big\|_1^2\right] \;\leq\; \widetilde{C}_0,$$

with $\widetilde{C}_0 = C_*(\alpha_\kappa)\beta$ up to a constant depending on $\alpha_\kappa$ and the covering radius used to select a nearest historical policy. Applying the Eluder argument to

$$\mathcal{G}_\Psi^{[\kappa]} := \left\{ (\pi,\tau_{h-1},o_h,a_h) \mapsto \sum_{t=h}^{\min(h+\kappa-1,\,H)} \big\|[G_t^{\Phi,\pi}(o_t,a_t) - G_t^{\Phi^\star,\pi}(o_t,a_t)]\, q_t^{\mathrm{mid},\xi^\star}\big\|_1^2 :\, \Phi\in\Psi\right\},$$

whose $\ell_2$-Eluder dimension is $d_{E,[\kappa]}^{(\Phi)}$, yields $\big|\mathcal{K}_{\text{adv}}^{\text{bad}}\big| \leq \widetilde{\mathcal{O}}\big(d_{E,[\kappa]}^{(\Phi)}\big)$. $\qquad\square$

### E.3 FINAL REGRET BOUND

We combine the preceding results by separating the contribution of *bad* and *good* batches. Recall that on data-collection rounds of batch $j$ the realized adversary is stationary (Remark 4), so the per-round instantaneous regret equals

$$\underbrace{V^{\pi_j,\,\mathcal{R}(\pi_j)}(\xi_j)}_{\text{optimistic value}} - \underbrace{V^{\pi_j,\,\mathcal{R}_{\xi^\star}(\pi_j)}(\xi^\star)}_{\text{true value}}.$$

**Lemma 11** (Regret on Bad Batches). *Let $\mathcal{K}^{\text{bad}}$ be the set of bad batches. Then*

$$\sum_{j\in\mathcal{K}^{\text{bad}}}\sum_{t\in\mathrm{Batch}_j} \left(V^{\pi_j,\,\mathcal{R}(\pi_j)}(\xi_j) - V^{\pi_j,\,\mathcal{R}_{\xi^\star}(\pi_j)}(\xi^\star)\right) \;\leq\; \widetilde{\mathcal{O}}\!\left((d_E^{(\theta)} + d_E^{(\Phi)})\cdot\frac{T}{K}\cdot H\right).$$

*Proof.* By Proposition 4, $|\mathcal{K}^{\text{bad}}| \leq \widetilde{\mathcal{O}}(d_{E,[\kappa]}^{(\theta)} + d_{E,[\kappa]}^{(\Phi)})$. For any batch $j$ and any data-collection round $t \in \mathrm{Batch}_j$, Lemma 5 with rewards in $[0,1]$ implies $V^{\pi_j,\,\mathcal{R}(\pi_j)}(\xi_j) - V^{\pi_j,\,\mathcal{R}_{\xi^\star}(\pi_j)}(\xi^\star) \leq H\big\|\mathbb{P}_{\xi_j}^{\pi_j,\mathcal{R}(\pi_j)} - \mathbb{P}_{\xi^\star}^{\pi_j,\mathcal{R}_{\xi^\star}(\pi_j)}\big\|_1 \leq 2H$. Hence the regret of one bad batch is at most $2H \cdot |\mathrm{Batch}_j|$, and with $|\mathrm{Batch}_j| \asymp T/K$ the stated bound follows. $\qquad\square$

**Lemma 12** (Regret on Good Batches). *Let $\mathcal{K}^{\mathrm{good}}$ be the complement of $\mathcal{K}^{\mathrm{bad}}$. Then*

$$\sum_{j \in \mathcal{K}^{\mathrm{good}}} \sum_{t \in \mathrm{Batch}_j} \left( V^{\pi_j,\, \mathcal{R}(\pi_j)}(\xi_j) - V^{\pi_j,\, \mathcal{R}_{\xi^\star}(\pi_j)}(\xi^\star) \right) \leq \widetilde{\mathcal{O}}\left( H \sqrt{(d_{E,[\kappa]}^{(\theta)} + d_{E,[\kappa]}^{(\Phi)})\, T\, C(\alpha_\kappa)\, \beta} \right).$$

*Proof.* Fix a good batch $j$. By Lemma 5 and Lemma 6, the per-round regret is bounded by $H$ times the same-policy total variation, which further splits into a *world* part and an *adversary* part (plus an initial-state term that is accounted for identically). Summing linearly over all data-collection rounds in good batches and applying Cauchy–Schwarz yields:

$$\sum_{j \in \mathcal{K}^{\mathrm{good}}} \sum_{t \in \mathrm{Batch}_j} \mathrm{WorldError}_t \leq \sqrt{T} \left( \sum_{j \in \mathcal{K}^{\mathrm{good}}} \mathcal{E}_{\mathrm{world}}(j) \right)^{1/2},$$

$$\sum_{j \in \mathcal{K}^{\mathrm{good}}} \sum_{t \in \mathrm{Batch}_j} \mathrm{AdvError}_t \leq \sqrt{T} \left( \sum_{j \in \mathcal{K}^{\mathrm{good}}} \mathcal{E}_{\mathrm{adv}}(j) \right)^{1/2}.$$

For a good batch $j$, definition of bad batch ensures that the optimistic model $\xi_j$ has small in-batch squared errors relative to the historical fit precondition from Proposition 3. As in **?**, this lets us apply the $\ell_2$-type Eluder counting principle over the sequence of *good-batch* rounds, giving

$$\sum_{j \in \mathcal{K}^{\mathrm{good}}} \mathcal{E}_{\mathrm{world}}(j) \leq \widetilde{\mathcal{O}}\big(d_{E,[\kappa]}^{(\theta)} C(\alpha_\kappa)\, \beta\big), \qquad \sum_{j \in \mathcal{K}^{\mathrm{good}}} \mathcal{E}_{\mathrm{adv}}(j) \leq \widetilde{\mathcal{O}}\big(d_{E,[\kappa]}^{(\Phi)} C(\alpha_\kappa)\, \beta\big).$$

Combining the two components and multiplying by the factor $H$ from Lemma 5 yields the claim. $\square$

*Proof of Theorem 1.* Let $d_E := d_{E,[\kappa]}^{(\theta)} + d_{E,[\kappa]}^{(\Phi)}$. By Lemma 11 and Lemma 12, and adding the warm-up cost $H(m-1)K$ (Remark 4),

$$PR(T) \leq \underbrace{\widetilde{\mathcal{O}}\big(H\sqrt{d_E\, T\, \beta}\big)}_{\text{good batches}} + \underbrace{\widetilde{\mathcal{O}}\big(H\, d_E\, T/K\big)}_{\text{bad batches}} + \underbrace{H(m-1)K}_{\text{warm-up}}.$$

The first term is $K$-independent. Balancing the second and third terms by $K = \left\lceil \sqrt{d_{E,[\kappa]}\, T} \right\rceil$ yields a combined contribution $\widetilde{\mathcal{O}}\big(H\, m\, \sqrt{d_{E,[\kappa]}\, T}\big)$, which together with the good-batch term gives the stated regret bound. $\square$

# F  SUPPLEMENTARY EXPLANATIONS FOR ASSUMPTIONS AND LEMMAS

## F.1  EXPLAINATIONS FOR POSTERIOR-LIPSCHITZ

In this part, we will present a counterexample showing that in multi-step weakly revealing dynamics, together with a bounded-memory and stationary opponent (without Posterior-Lipschitz), do not suffice to guarantee sublinear policy regret in POMGs.

**Theorem 2** (Counterexample under multi-step weakly revealing). *Fix a horizon $H \geq 2$ and an action-set size $|\mathcal{A}| = A \geq 2$. There exists a two-player zero-sum POMG such that:*

*(i) The adversary is stationary and $1$-memory, but not posterior-Lipschitz.*

*(ii) The world is $(\kappa{=}2, \alpha{=}1)$-weakly revealing.*

*(iii) For any learning algorithm and any $T \in \mathbb{N}^+$, if the instance is drawn uniformly at random from a finite family, then with probability at least $1/2$ over the instance draw,*

$$\mathrm{PR}(T) \geq \tfrac{1}{2} \min\{ A^{H-2},\, T \}.$$

*Proof.* **World.** Let $\mathcal{S} = \{s_1, s_2, s_3, s_4\}$, $|\mathcal{A}| = A \geq 2$, and $\mathcal{B} = \{b_{\text{coop}}, b_{\text{punish}}\}$. The observation alphabet is $\mathcal{O} = \{o_0, o^+, o^-\}$. For all $h \in [H]$,

$$E_h(\cdot \mid s_1) = E_h(\cdot \mid s_2) = \delta_{o_0}, \qquad E_h(\cdot \mid s_3) = \delta_{o^+}, \qquad E_h(\cdot \mid s_4) = \delta_{o^-}.$$

Thus $s_1, s_2$ are aliased in one step, while $s_3, s_4$ are distinguishable. The initial state is $s_1$. The learner receives reward 1 iff $s_H = s_3$, and 0 otherwise.

Transitions are controlled only by the adversary's action; for any $a \in \mathcal{A}$,

$$T_h(\cdot \mid s_1, a, b_{\text{coop}}) = \delta_{s_3}, \quad T_h(\cdot \mid s_2, a, b_{\text{coop}}) = \delta_{s_4},$$
$$T_h(\cdot \mid s_3, a, b_{\text{coop}}) = \delta_{s_3}, \quad T_h(\cdot \mid s_4, a, b_{\text{coop}}) = \delta_{s_4},$$
$$T_h(\cdot \mid s, a, b_{\text{punish}}) = \delta_{s_4} \quad \text{for all } s \in \mathcal{S}.$$

For any $h$, consider the two-step emission matrix $M_h^{(2)}$ with rows indexed by $(o_h, o_{h+1})$ and columns by $s \in \mathcal{S}$. Under $b_{\text{coop}}$, the two-step sequences are

$$s_1 \mapsto (o_0, o^+), \quad s_2 \mapsto (o_0, o^-), \quad s_3 \mapsto (o^+, o^+), \quad s_4 \mapsto (o^-, o^-),$$

which yield a $4 \times 4$ identity submatrix of $M_h^{(2)}$. Hence $M_h^{(2)}$ has full column rank and $\sigma_{\min}(M_h^{(2)}) = 1$, so the world is ($\kappa=2, \alpha=1$)-weakly revealing.

**Adversary.** Let $\Pi$ denote the learner's policy class. Consider deterministic "open-loop" policies that fix the first $H - 2$ actions and are arbitrary afterwards:

$$\bar{\Pi} := \left\{ \pi^{(u)} : \pi_h^{(u)}(\cdot \mid \tau_A) = \delta_{u_h} \text{ for all reachable } \tau_A, \ 1 \leq h \leq H - 2 \right\},$$

where $u = (u_1, \ldots, u_{H-2}) \in \mathcal{A}^{H-2}$. There are $M := |\bar{\Pi}| = A^{H-2}$ such policies.

Draw a "secret" sequence $u^\star$ uniformly from $\mathcal{A}^{H-2}$, let $\pi^\star \in \bar{\Pi}$ be the corresponding open-loop policy, and let $\Pi(u^\star) \subseteq \Pi$ be the set of policies that are behaviorally equivalent to $\pi^\star$ on all reachable histories when the adversary plays $b_{\text{coop}}$ in all steps.

Define a stationary response map $g : \Pi \to \Psi$ by

$$g(\pi) = \begin{cases} \mu_{\text{coop}}, & \pi \in \Pi(u^\star), \\ \mu_{\text{punish}}, & \text{otherwise}, \end{cases}$$

where $\mu_{\text{coop}}$ ($\mu_{\text{punish}}$) plays $b_{\text{coop}}$ ($b_{\text{punish}}$) at every stage. Across episodes we set $R_t(\pi^{1:t}) := g(\pi_t)$. Then: $R_t$ depends only on $\pi_t$ (so the adversary is 1-memory), the same $g$ is used for all $t$ (stationary) .

On the other hand, $g$ is not Posterior-Lipschitz. Indeed, fix any $h$ and consider any $\tau_{B,h} = (o_{B,1}, b_{\text{coop}}, o_{B,2}, b_{\text{punish}}, o_{B,3}, b_{\text{coop}}, \ldots)$, where the action ($b_{\text{coop}}$) and $b_{\text{punish}}$ are alternated throughout the episode and $(o_{B,1}, \ldots, o_{B,h})$ is any feasible sequence of observations under the alternate action sequences ($b_{\text{coop}}, b_{\text{punish}}, \ldots,$). Since $\tau_{B,h}$ above does not correspond to the scenario that the adversary plays $b_{\text{coop}}$ in all steps, there exist two policies $\pi \in \Pi(u^*)$ and $\nu \notin \Pi(u^*)$ such that $\pi(\cdot|\tau_{A,h}) = \nu(\cdot|\tau_{A,h})$, for all $\tau_{A,h}$ such that $\Pr(\tau_{A,h}|\tau_{B,h}) > 0$. This clearly violates the Posterior-Lipschitz condition.

**Regret lower bound.** For any $\pi \in \Pi$, letting $V(\pi, \mu)$ be the value under stationary adversary policy $\mu$, the transition structure implies

$$V(\pi, \mu_{\text{coop}}) = 1, \qquad V(\pi, \mu_{\text{punish}}) = 0.$$

Fix the instance $u^\star$ and take comparator $\pi^\star \in \Pi(u^\star)$. For every episode $t$,

$$V(\pi^\star, R_t([\pi^\star]_t)) = 1, \qquad V(\pi_t, R_t(\pi^{1:t})) = \mathbf{1}\{\pi_t \in \Pi(u^\star)\}.$$

Hence

$$\text{PR}(T) \geq \sum_{t=1}^{T} \left(1 - \mathbf{1}\{\pi_t \in \Pi(u^\star)\}\right) = \sum_{t=1}^{T} \mathbf{1}\{\pi_t \notin \Pi(u^\star)\}.$$

We now bound from below the number of episodes with $\pi_t \notin \Pi(u^\star)$ when $u^\star$ is drawn uniformly. For any fixed $\pi \in \Pi$, there is at most one sequence $u^\star$ such that $\pi \in \Pi(u^\star)$, so

$$\Pr_{u^\star}\left(\pi \in \Pi(u^\star)\right) \leq \tfrac{1}{M}, \qquad M = A^{H-2}.$$

Moreover, as long as $\pi_1, \ldots, \pi_{t-1} \notin \Pi(u^\star)$, the adversary plays $\mu_{\text{punish}}$ in episodes $1, \ldots, t-1$. Under this policy the state is sent to $s_4$ at the first step and remains there, so the entire history consists of the same observation stream $(o_0, o^-, \ldots, o^-)$, independent of $u^\star$. Thus, conditional on $\{\pi_1, \ldots, \pi_{t-1} \notin \Pi(u^\star)\}$, the instance $u^\star$ is still uniform and independent of $\pi_t$, and

$$\Pr\left(\pi_t \in \Pi(u^\star) \,\middle|\, \pi_1, \ldots, \pi_{t-1} \notin \Pi(u^\star)\right) \leq \tfrac{1}{M}.$$

Let $T' := \min\{T, M/2\}$. By a union bound,

$$\Pr\left(\exists t \leq T' : \pi_t \in \Pi(u^\star)\right) \leq \tfrac{T'}{M} \leq \tfrac{1}{2},$$

so

$$\Pr\left(\forall t \leq T' : \pi_t \notin \Pi(u^\star)\right) \geq \tfrac{1}{2}.$$

On this event,

$$\mathrm{PR}(T) \geq \sum_{t=1}^{T'} \mathbf{1}\{\pi_t \notin \Pi(u^\star)\} = T' \geq \tfrac{1}{2}\min\{T, M\} = \tfrac{1}{2}\min\{T, A^{H-2}\},$$

which proves the claim.

$\square$

### F.2 PROOFS OF SUPPORTING LEMMAS

*Proof of Lemma 2.* We prove the adversary case and the world case follows by the substitutions $(B, d_{\text{adv}}, \Phi^\star, w) \mapsto (O_A, d_w, W^\star, u)$.

By Example 1, for any history $\tau$ the adversary response is linear: $g(x) = \Phi^\star w(\tau)$, where the operator $\Phi^\star \in \mathbb{R}^{B \times d_{\text{adv}}}$ is unknown and the weights $w(\tau) \in \mathbb{R}^{d_{\text{adv}}}$ are bounded (e.g., $w(\tau) \in \mathbb{R}^{d_{\text{adv}}}$, so $\|w(x)\|_2 \leq 1$).

Fix any linear reparameterization that collects exactly the free entries of $\Phi^\star$ into a vector $\theta \in \mathbb{R}^d$ with $d = B\, d_{\text{adv}}$. Write this as $\theta = \mathrm{vec}(\Phi^\star)$ for some $\Phi^\star \in \mathbb{R}^{B \times d_{\text{adv}}}$. For each coordinate $i \in \{1, \ldots, B\}$, define the feature map

$$\varphi(\tau, i) := e_i \otimes w(\tau) \in \mathbb{R}^d, \qquad \|\varphi(\tau, i)\|_2 = \|w(\tau)\|_2 \leq 1,$$

and the corresponding scalar output $y_i(x) := e_i^\top g(x)$. By construction,

$$y_i(\tau) = e_i^\top g(\tau) = e_i^\top \tilde{\Phi}^\star w(\tau) = \left(e_i^\top \otimes w(\tau)^\top\right) \mathrm{vec}(\tilde{\Phi}^\star) = \langle \varphi(\tau, i), \theta \rangle.$$

Hence each coordinate belongs to a $d$-parameter linear class with bounded features. By Example 4 of Russo & Van Roy (2013), the $\varepsilon$-eluder dimension of such a class is $\mathcal{O}\big(d \log(1/\varepsilon)\big)$. Absorbing logarithmic factors into $\widetilde{\mathcal{O}}(\cdot)$ yields $\dim_E(\mathcal{G}_\Phi^{[\kappa]}) = \widetilde{\mathcal{O}}(d_{\text{adv}} B)$. The same argument with $(O_A, d_w, W^\star, u)$ gives $\dim_E(\mathcal{F}_\theta^{[\kappa]}) = \widetilde{\mathcal{O}}(d_w O_A^\kappa)$.

$\square$

