# OpenReview forum: "Policy Regret Minimization in Partially Observable Markov Games"
_ICLR.cc/2026/Conference — ICLR 2026 Conference Withdrawn Submission_

### Official Review · Reviewer_9NX7 · 2025-10-31

**Soundness:** 3
**Presentation:** 2
**Contribution:** 2
**Rating:** 4
**Confidence:** 3

**Summary:**

This paper proposes the first unified framework for policy regret minimization in partially observable Markov games (POMGs) against adaptive opponents. The authors achieve sublinear policy regret under bounded-memory and weakly revealing conditions. Some new techniques, such as the Posterior-Lipschitz assumption and operator decomposition for POMGs are provided.

**Strengths:**

1. The paper provides a theoretical policy regret analysis about POMGs. The theoretical analysis is solid and comprehensive.

2. The algorithm in this paper successfully combines the Optimistic MLE in [1] for POMDP, some assumptions and techniques in [2] for policy regret minimization, and batch analysis in [3] for policy low-switching. The final algorithm successfully solves Partially Observed Markov Games.

3. The paper is well-structured. It also contains some sketches for the Appendix. The algorithms are presented in a clear way.

[1]. Qinghua Liu, Alan Chung, Csaba Szepesva ́ri, and Chi Jin. When is partially observable reinforcement learning not scary?

[2]. Thanh Nguyen-Tang and Raman Arora. Learning in markov games with adaptive adversaries: Policy regret, fundamental barriers, and efficient algorithms.

[3]. Nuoya Xiong, Zhaoran Wang, and Zhuoran Yang. A general framework for sequential decisionmaking under adaptivity constraints.

**Weaknesses:**

1. Although the paper is technically dense, its methodological novelty appears limited. The main proof largely combines the OMLE techniques from [1] with the policy regret algorithm from [2], without introducing substantial new technical contributions. The main difference between OMLE and this paper is that it contains the class of adversarial channel $g$ in the MLE oracle. However, it will not introduce intrinsic difficulty since MLE analysis still works.

2. The proposed algorithm seems difficult to implement in practice. It involves solving a constrained optimization problem whose structure does not appear to lend itself to tractable solution methods. The algorithm also contains the adversarial channel in the MLE oracle, which can make this constraint optimization harder to solve.

3. Several technical definitions are introduced without sufficient motivation or intuitive explanation. For example, Section 4.3 presents the definitions of the eluder dimension and the function class directly, which may be challenging for readers unfamiliar with prior work on POMDPs and RL theory. Providing intuitive explanations or brief context for these definitions would improve readability and accessibility.


[1]. Qinghua Liu, Alan Chung, Csaba Szepesva ́ri, and Chi Jin. When is partially observable reinforcement learning not scary?

[2]. Thanh Nguyen-Tang and Raman Arora. Learning in markov games with adaptive adversaries: Policy regret, fundamental barriers, and efficient algorithms.

**Questions:**

1. Could the author explain why the Posterior-Lipschitz assumption is necessary? It seems that this assumption is used in Lemma 7 to get an upper bound $\Delta_\sigma(\pi,\upsilon)$, which then reappears in Lemma 10 as an upper bound term. After that, the paper seems to treat this term as a constant. Then, why the Posterior-Lipschitz assumption necessary? It would be helpful if the authors could explain the role and necessity of this assumption more explicitly.

2. In Line 216, is it correct that the conditional distribution of $\tau_A$ given $\tau_B$ should also depend on the policy of both agent and the adversarial?

3. Some notations like $g(\cdot \mid \tau_B, \pi^{1:m})$ should be clarified before they are used. The paper only defines $g$ as a function that maps $\Pi^M$ to the adversarial policy space.

4. Do the authors have any insights or suggestions on how to solve the constrained optimization problem in practice? Are there some potential empirical applications for this paper?

---

> ### Author Response · Authors · 2025-11-19
>
> We thank the reviewer for the thoughtful comments and suggestions.
>
> > Weakness 1
>
> We respectfully disagree that our proof merely combines the OMLE techniques
> from [1] with the policy-regret algorithm from [2], or that adding adversarial
> channels to the MLE oracle “does not introduce intrinsic difficulty.” While we
> build on these frameworks, the Stackelberg POMG setting requires several new
> technical ideas.
>
> First, our setting is structurally different from both [1] and [2]. Work [1]
> treats a single-agent POMDP with a passive environment, and [2] analyzes policy
> regret in fully observable Markov games. In our partially observable
> Stackelberg POMG, however, the learner’s trajectory aggregates the effects of
> the unknown world model $\theta$ and the adversarial channel $\Phi$, so the
> uncertainty about these two components is intrinsically coupled. As a result,
> maintaining two independent confidence sets for $\theta$ and $\Phi$ in the
> style of [1] is statistically unsound: any plausible model for $\theta$
> restricts the admissible $\Phi$, and vice versa. Our Algorithm~1 therefore
> works with a single confidence region $\mathcal C \subseteq \Xi$ over the
> joint parameter $\xi=(\theta,\Phi)$ and requires a new analysis of the
> corresponding joint MLE.
>
> Second, handling adversarial channels and policy regret is not “just one more
> parameter on top of OMLE.” In our setting, each per-step observable operator
> $J_h^{\xi,\pi}$ is a coupled black box depending on both the joint parameter
> $\xi=(\theta,\Phi)$ and the learner’s policy block $\pi$, a regime outside
> the scope of the OOM analysis in [1]. We develop a causal decomposition that
> separates this operator into a world channel $W^\theta$ and an adversary
> channel $G^{\Phi,\pi}$, and impose Posterior-Lipschitz to control how
> $G^{\Phi,\pi}$ can change with the adversary’s posterior. This structure
> yields joint-likelihood bounds for the coupled $(\theta,G^{\Phi,\pi})$ system,
> which are then used to control the value error of optimistic joint models in
> our blockwise policy-regret analysis. Such a tight coupling between OOM-based
> estimation and policy-regret optimization does not appear in either [1] or [2].
>
> In our revised version, we have updated the overview-of-techniques section to
> make these distinctions relative to [1] and [2] more explicit.
>
> > Weakness 2
>
> As we deal with general function approximation for the adversary behavior, solving the constrained optimization is computationally challenging, as is typical in RL theory work with general function approximation. However, for several common function classes—such as linear models (e.g., Example 1 in our paper)—the constrained optimization can admit efficient, closed-form solutions.
>
> > Weakness 3
>
> We appreciate this comment and agree that these definitions are quite technical for readers who are not yet familiar with eluder-based RL theory. In the revised version, we have improved the exposition in Section 4.3 in two ways.
>
> First, we expanded the introductory paragraph to explain that our batched analysis relies on an $\ell_2$-type Eluder condition and we now explicitly point readers to
> [3, Sec. 5.1] for additional background, examples, and a comparison with the standard Eluder dimension.
>
> Second, immediately after Definition 3 we added an intuitive sentence
> summarizing the meaning of the definition, explaining that if functions in the
> class fit the past data well on average, then large squared errors can occur on
> at most $O(d\log T)$ rounds, so the total prediction error is controlled by the
> Eluder dimension.
>
> We hope these additions make the technical definitions in Section 4.3 more
> accessible to readers who are less familiar with eluder-based RL theory.
>
> [3]. Nguyen-Tang, Thanh, and Raman Arora. "Policy-Regret Minimization in Markov Games with Function Approximation." In Forty-second International Conference on Machine Learning.

---

> ### Author Response · Authors · 2025-11-19
>
> > Question 1
>
> As discussed in our response to Reviewer 6LiP, we construct a
> counterexample in Appendix F.1 to show that the Posterior-Lipschitz assumption is
> information-theoretically necessary. We hope this also addresses your concern.
>
> Beyond this counterexample, we use a posterior-based formulation because the
> adversary never observes the learner’s private trajectory $\tau_A$. It only
> has access to its own history $\tau_B$ and thus to a posterior belief about
> how the learner will act. Any reasonable response rule can therefore be viewed
> as a mapping from this posterior to a mixed action. Assumption 1.3 simply
> imposes a Lipschitz regularity on this mapping: if two policy blocks
> induce similar posterior behavior from the adversary’s viewpoint, then the
> adversary’s response distributions cannot differ arbitrarily.
>
> In the proof, Assumption 1.3 is first used in Lemma 7 to control the variation
> of the adversary channel $G^{\Phi,\pi}$ when we compare two policy blocks
> $\pi_{1:m}$ and $\nu_{1:m}$. Posterior-Lipschitz ensures that,
> whenever the induced posteriors are close, the resulting response distributions
> are also close in total variation. This yields a uniform bound
> $\Delta_\sigma(\pi,\nu)$ on the difference between the two adversary
> channels. In other words, Assumption 1.3 prevents small changes in the
> adversary’s posterior from causing arbitrarily large changes in
> $G^{\Phi,\pi}$.
>
> Lemma 10 then reuses $\Delta_\sigma(\pi,\nu)$ as an upper-bound term in a
> likelihood-ratio argument for the joint trajectory law. Once Assumption 1.3
> guarantees a uniform finite bound $\Delta_\sigma(\pi,\nu) \le C_\sigma$ for
> all feasible policy blocks $\pi,\nu$, this constant $C_\sigma$ can be
> absorbed into the problem-dependent prefactor in our final $O(\sqrt{T})$
> regret bound. The fact that $\Delta_\sigma(\pi,\nu)$ appears only through
> such a constant does not mean that Assumption 1.3 is superfluous: without this
> regularity, $\Delta_\sigma(\pi,\nu)$ can be arbitrarily large even when the
> adversary’s posterior is nearly unchanged (as in our “needle in a haystack”
> construction). In such cases, the likelihood control in Lemmas 7 and 10 would
> break down, and as a result, any finite sublinear policy-regret bound would
> also fail.
>
> > Question 2
>
> Yes. In fact, the conditional
> distribution of $\tau_A$ given $\tau_B$ is defined under the joint trajectory
> law induced by the world model and both players' policies.
>
> In Assumption 1, given the true world model $\theta$, a learner policy
> block $\pi_{1:m}$, and the induced adversary response rule $g$, we denote
> by $P^{\pi_{1:m},g,\theta}(\tau_A,\tau_B)$ the resulting joint trajectory
> distribution.
> The conditional expectation in Line 216 should therefore be read as
> $S_{\tau_B}(\pi_h^i) := E_{\tau_A \sim P^{\pi_{1:m},g,\theta}(\cdot \mid \tau_B)}\left[ \pi_h^i(\cdot \mid \tau_A) \right].$
>
> In the revised version, we have revised the definition of Assumption 1.3 in blue
> to make this dependence explicit and to avoid confusion.
>
> > Question 3
>
> Thank you for pointing this out. In the revised version, we clarify both the
> joint trajectory law and the induced adversarial policy. Specifically, in
> Assumption 1.3 (Posterior-Lipschitz), we now write in blue:
>
> Let $g(\cdot \mid \tau_B,\pi^{1:m})_h$ denote the distribution of the
> adversary's action at step $h$ induced by the response rule $g$ given
> private trajectory $\tau_B$ and learner policies $\pi^{1:m}$. Then there
> exists a constant $L \ge 0$ such that, for any two policy blocks
> $\pi^{1:m},\nu^{1:m}$, any $h \in [H]$, and any adversary trajectory
> $\tau_B$, $ g(\cdot \mid \tau_B,\pi^{1:m})_h - g(\cdot \mid \tau_B,\nu^{1:m})_h \le$
>
> $L \max_{i\in[m]} \lVert S_{\tau_B}(\pi_h^i) - S_{\tau_B}(\nu_h^i)\rVert.$
>
> In the revised version, we have clarified the definition of $g$ before using
> it in Assumption 1.3. We hope this revised formulation will address your
> Questions 2 and 3 and clarify the definition.
>
> > Question 4
>
> As we deal with general function approximation for the adversary behavior, solving the constrained optimization is computationally challenging, as is typical in RL theory work with general function approximation. However, for several common function classes—such as linear models (e.g., Example 1 in our paper)—the constrained optimization can admit efficient, closed-form solutions. Regarding the empirical applications for this paper, our framework is directly motivated by the optimal taxation problem in the Economist AI framework, thus can be applied to this domain. Please see our response to Weakness 1 of Reviewer 6LiP.

---

> ### Author Response · Authors · 2025-11-25
>
> Dear our Reviewer,
>
> Thank you very much for taking the time to read and consider our rebuttal. We hope our response has adequately addressed your concerns. If so, we would greatly appreciate any updates to your review and rating wherever you see fit.
>
> If there are remaining questions or issues, we would be more than happy to clarify them during the remaining author–reviewer discussion period.
>
> Thank you again for your time and effort.
>
> Best regards,
>
> Authors

---

### Official Review · Reviewer_6LiP · 2025-11-01

**Soundness:** 2
**Presentation:** 2
**Contribution:** 2
**Rating:** 4
**Confidence:** 4

**Summary:**

The paper studies policy-regret minimization for a learner playing a partially observable Markov game (POMG) against an adaptive, bounded-memory, stationary opponent. The authors propose a batched, model-based algorithm (MOMLE) that maintains a single joint confidence set over the world model and the opponent’s response model via optimistic MLE on the learner-observable process. A key technical ingredient is an OOM-based “causal decomposition” of per-step operators into a world channel and an adversary channel, enabling a telescoping analysis. Under multi-step weakly revealing observations and a posterior-Lipschitz opponent, the method achieves a sublinear policy-regret bound.

**Strengths:**

1. First policy-regret result in POMGs (as far as I know). Extends recent MG results to imperfect information with adaptive opponents.
2. The OOM factorization into world/adversary channels plus a two-stage telescoping bound is technically interesting and seems reusable.

**Weaknesses:**

1. Although the policy regret setup makes sense in general, it makes less sense in POMG, which features decentralized information. Specifically, how can the adversary response map depend on the learner's past policies since such learner's information is almost never available to the adversary in a decentralized setup.

2. Although I understand Assumption 1.1&1.2 is necessary for tractable algorithms, it again makes less sense to me. In my opinion, it is almost saying that the opponent is ``stationary''. This kind of defeats the purposes of considering adaptive opponents. In fact, the policy regret considered in this paper is much weaker than the standard external regret which allows adversarial opponents. As a side note, the regret considered by liu et al, 2022 is not by definition external regret in my opinion. It is a self-play setting where both players are controlled by a certain algorithm. For the actual external regret guarantee, plz refer to [1, 2], which is in most cases hard. This further raises questions regarding how interesting it is to study policy regret.

3. Based on the intuition that Assumption 1.1&1.2 make the opponent almost stationary, it is kind of expected that the problem reduces to a single-agent POMDP (up to extensions on the state space to incorporate the finite memory dependence of the adversary).

4. Assumption 1.3 also lacks justifications and requires explanations. It is unclear whether it is fundamental or only makes analysis possible.

[1]. Liu, Qinghua, Yuanhao Wang, and Chi Jin. "Learning markov games with adversarial opponents: Efficient algorithms and fundamental limits." International Conference on Machine Learning. PMLR, 2022.

[2]. Foster, Dylan J., Noah Golowich, and Sham M. Kakade. "Hardness of independent learning and sparse equilibrium computation in markov games." International Conference on Machine Learning. PMLR, 2023.

**Questions:**

See above

---

> ### Author Response · Authors · 2025-11-19
>
> We thank the reviewer for the constructive feedback.
>
> ---
>
> > Weakness 1
>
> Policy-regret learning is, in fact, particularly well-suited to partially observable environments which prevail in mechanism design settings. Indeed, our problem setup is directly motivated by the optimal taxation problem studied in the AI Economist (Zheng et al., 2020), which we discussed in the second paragraph of our Introduction. In this setting, a government agent (the learner) aims to learn an optimal tax schedule T(z) based on agents’ incomes z over tax periods lasting M episodes (corresponding to the mini-batch size T/K in our algorithm).
>
> The adaptive adversaries in our framework correspond to the adaptive economic agents who gather resources, build houses, and trade. *Their behaviors are governed by a response function that depends on the learner’s past policies — namely, the tax schedules from the current and previous years*. Furthermore, *the optimal taxation problem in  (Zheng et al., 2020) is  inherently a partially observable Markov game*, as the learner cannot observe private information of the agents, such as their skills or labors. We refer the reviewer to the AI Economist (Zheng et al., 2020) for more details about their setup.
>
> ---
> > Weakness 2
>
> - Regarding the **stationary** point: We would like to clarify that “stationary” in our assumptions does not undermine the role of adaptivity. The adversary is stationary only in the temporal dimension, not in its responsiveness. It still adapts to the entire sequence of the learner’s past policies and, in fact, has unlimited computational power to respond to any such sequence within its memory. In addition, this form of stationarity is known to be necessary for policy-regret learning [Nguyen-Tang \& Arora (2024)]
>
> - Regarding **"policy regret
> vs the standard external regret"**: In fact, it is quite the opposite: policy regret is a strict generalization of external regret in (PO)MGs. When the adversary’s memory size m = 0, the adversary becomes oblivious (yet potentially adversaral), and policy regret reduces exactly to external regret. Moreover, none of our assumptions on the adaptive adversaries restrict them from being adversarial; the framework fully allows adversarial behavior as well as other adaptive behaviors such as strategic ones.
>
> - Regarding Liu et al, 2022: First, we agree. The citation to “Liu et al., 2022b” on the second page regarding "external regret minimization" was a mistaken reference. It should have referred to [1], and we have corrected this in the revised version. Second, we would like to re-iterate the motivation of policy regret learning, in the context of self-play setting in Liu et al. 2022b and of external regret learning. Most MARL research has focused on learning optimal strategies under the assumption that
> agents have no incentive to deviate unilaterally. This motivates the use of solution concepts such
> as Nash, correlated, or coarse correlated equilibrium in self-play and decentralized settings [Liu et al, 2022], and
> external regret in online learning. External regret compares the learner’s performance to the best
> fixed strategy in hindsight, assuming the opponents’ responses remain fixed. However, this assumption overlooks many real-world settings—e.g., Stackelberg games, auction design, and mechanism
> design—where agents are adaptive and respond strategically to others’ deviations. Consequently,
> one agent’s decision to deviate must account for how others will adapt in response. Our paper fills in this gap in the literature.
> ---
> > Weakness 3
>
> This statement oversimplifies almost any online learning formulation in MARL, including the external policy-regret setting against an oblivious sequence of opponent policies studied in [1]. In online learning, the interaction, and thus the learning problem, is indeed often reduced to a “single-agent” viewpoint where we control only one learner and treat the remaining agents as part of the (uncontrolled) environment — whether they are oblivious, as in [1], or more generally adaptive, as in our work. However, the central challenge remains the same in both [1] and our setting (even under Assumptions 1.1 and 1.2): the learner effectively faces different environments when executing different policies. This non-stationarity induced by policy-dependent environment transitions is what makes the learning problem difficult in both frameworks.
>
> ---
> > Weakness 4
>
> Assumption 1.3 is fundamental for our settings. In our revised
> version we have added a counterexample in Appendix F.1 to show that
> in multi-step weakly revealing dynamics, together with a bounded-memory
> and stationary adversary (without Posterior-Lipschitz), do not suffice to
> guarantee sublinear policy regret in POMGs. If the adversary’s response
> map is allowed to be highly discontinuous (violating Assumption 1.3), then
> any algorithm can suffer *linear or exponential* policy regret.

---

> > ### Author Response · Authors · 2025-11-25
> >
> > Dear our Reviewer,
> >
> > Thank you very much for taking the time to read and consider our rebuttal. We hope our response has adequately addressed your concerns. If so, we would greatly appreciate any updates to your review and rating wherever you see fit.
> >
> > If there are remaining questions or issues, we would be more than happy to clarify them during the remaining author–reviewer discussion period.
> >
> > Thank you again for your time and effort.
> >
> >
> > Best regards,
> >
> > Authors

---

### Official Review · Reviewer_eex2 · 2025-11-03

**Soundness:** 4
**Presentation:** 4
**Contribution:** 4
**Rating:** 10
**Confidence:** 2

**Summary:**

This paper studies the problem of minimizing policy regret against an adaptive adversary in Markovian Games with partial observability. As noted by this paper, this is a technically challenging problem at multiple levels, with policy regret even under full observability requiring structural conditions for sample efficient learning. The main contribution is in identifying reasonable conditions under which policy regret minimization is possible in POMG — specifically restrictions on the nature of the adaptive adversary - i.e. memory bound, stationary over time and a novel assumption about posterior-Lipschitzness, which is a condition about the stability of the adversary’s repsonses to different learner sequences that induce similar posterior beliefs about the learner’s future behavior. Additional assumptions are made about the nature of observability (required for the any reasonable inference about the hidden state) and the Eluder dimension, a complexity measure of the joint world state - adversary trajectories that can generate the observed states.  Under these assumptions, they provide an algorithmic result, building upon existing tools for POMGs and policy regret minimizing along with novel technical analysis to break through unique roadblocks due to the combination of an adaptive adversary and partial observability. Specifically, they adapt the OOM framework of Liu et al. after using a novel technical tool to causally disambiguate the world state from the adversary actions.

**Strengths:**

This paper solves a genuinely difficult problem, stitching together tools for two problems with different sources of complexity -- partial observability and policy regret. Based on my limited experience in this field, this result appears to be a significant technical advancement of the field and is worthy of acceptance on those grounds.

**Weaknesses:**

NA

**Questions:**

Is there any interplay between the assumptions about the adaptive adversary and about the nature of the observability and complexity of the POMG or are they uncoupled?

---

> ### Author Response · Authors · 2025-11-19
>
> We thank the reviewer for the positive assessment and for this insightful question.
>
> > Question 1
>
> Policy regret learning in POMGs inherits the hardness results from both policy regret learning in Markov games (Nguyen-Tang \& Arora 2024) and external regret learning and equilibrium learning in POMGs (Papadimitriou \& Tsitsiklis, 1987), as policy regret generalizes external regret. Thus, to study policy regret learnability in POMGs, it is natural to induce at least the sufficient conditions for policy regret learning (Bounded Memory, Stationarity, and Posterior-Lipschitz), and the sufficient conditions for external regret learning of POMGs (the multi-step weakly revealing condition). These conditions are absorbed into the dynamic model $\theta$ and the adversary model $\Phi$. However, the joint model $\xi=(\theta,\Phi)$ induces a coupled dynamic from the learner's perspective, in which the learner cannot distinguish whether the randomness and nonstationarity in its observations should be attributed to the environment dynamics or to the adversary's adaptive response. Nevertheless, our theoretical framework reveals that sublinear policy regret is achievable in this coupled dynamics.

---

### Author Response · Authors · 2025-11-19

We have updated the paper and highlighted the changes in blue, based on the current suggestions from our reviewers. We are happy to make further revisions based on our ongoing discussions with the reviewers.

The revisions consist of (i) expanding Section 1.1 (Overview of Techniques) to better explain the new tools used in our analysis, (ii) clarifying the definition of Posterior-Lipschitz in Section 3.1, (iii) adding an introduction and intuitive explanation of the Eluder condition in Section 4.3, and (iv) adding to Appendix F.1 a counterexample illustrating that, in the absence of Posterior-Lipschitz, multi-step weakly revealing dynamics alone are insufficient to guarantee sublinear policy regret.

---

### Author Response · Authors · 2025-11-28
**Last message**

Dear AC and Reviewers,

We recently received the announcement from ICLR 2026 stating that, due to the leaked-identity incident, reviewers will no longer participate in the post-rebuttal discussion. We fully respect this decision and understand that it may be the most appropriate course of action given the scale of submissions and the current circumstances.

However, we are concerned about its potential impact on individual submissions, including ours. We have made every effort to address and clarify all questions raised by the reviewers. Without the author–reviewer discussion phase, we are left uncertain as to whether our responses will be fully read or taken into account—especially since none of the concerns raised about our paper are critical, and most were clarification questions. We appreciate the reviewers’ comments, which have helped us improve the clarity of our paper, and we have directly and clearly addressed all issues in our rebuttal. We strongly believe that the two current ratings of 4 do not accurately reflect the contributions of our work. We would be happy to provide additional clarification if needed; however, the new policy prevents reviewers from engaging further with our responses or updating their evaluations accordingly.

We kindly ask the newly assigned AC and the reviewers to take our rebuttal into full consideration. As always, we are happy to provide any additional clarification (e.g., through AC communication) if there are remaining questions or concerns.

Best regards,
Authors

---

### Note · Authors · 2025-12-30

I have read and agree with the venue's withdrawal policy on behalf of myself and my co-authors.